# Obesity and disease severity magnify disturbed microbiome-immune interactions in asthma patients

David Michalovich [1], Noelia Rodriguez-Perez[2], Sylwia Smolinska [3,4], Michal Pirozynski [5], David Mayhew[6], Sorif Uddin[1,7], Stephanie Van Horn [8], Milena Sokolowska[2,9], Can Altunbulakli[2], Andrzej Eljaszewicz [2,9,10], Benoit Pugin[2], Weronika Barcik[2], Magdalena Kurnik-Lucka[11], Ken A. Saunders[1], Karen D. Simpson[1], Peter Schmid-Grendelmeier[9,12], Ruth Ferstl[2,9], Remo Frei[2,9], Noriane Sievi[13], Malcolm Kohler[13], Pawel Gajdanowicz [3,4], Katrine B. Graversen [14], Katrine Lindholm Bøgh [14], Marek Jutel[3,4], James R. Brown [6], Cezmi A. Akdis[2,9], Edith M. Hessel [1,16] & Liam O'Mahony[2,15,16]*

In order to improve targeted therapeutic approaches for asthma patients, insights into the molecular mechanisms that differentially contribute to disease phenotypes, such as obese asthmatics or severe asthmatics, are required. Here we report immunological and microbiome alterations in obese asthmatics (n = 50, mean age = 45), non-obese asthmatics (n = 53, mean age = 40), obese non-asthmatics (n = 51, mean age = 44) and their healthy counterparts (n = 48, mean age = 39). Obesity is associated with elevated proinflammatory signatures, which are enhanced in the presence of asthma. Similarly, obesity or asthma induced changes in the composition of the microbiota, while an additive effect is observed in obese asthma patients. Asthma disease severity is negatively correlated with fecal *Akkermansia muciniphila* levels. Administration of *A. muciniphila* to murine models significantly reduces airway hyper-reactivity and airway inflammation. Changes in immunological processes and microbiota composition are accentuated in obese asthma patients due to the additive effects of both disease states, while *A. muciniphila* may play a non-redundant role in patients with a severe asthma phenotype.

[1] Adaptive Immunity Research Unit, GSK R&D, Stevenage, UK. [2] Swiss Institute of Allergy and Asthma Research (SIAF), University of Zurich, Davos, Switzerland. [3] Department of Clinical Immunology, Wroclaw Medical University, Wroclaw, Poland. [4] ALL-MED' Medical Research Institute, Wroclaw, Poland. [5] Department of Allergology and Pulmonology, Centre of Postgraduate Medical Education, Warsaw, Poland. [6] Computational Biology, Human Genetics, GSK R&D, Collegeville, PA, USA. [7] Boehringer Ingelheim, 88397 Biberach an der Riß, Germany. [8] Target and Pathway Validation, Target Sciences, GSK R&D, Collegeville, PA, USA. [9] Christine Kühne-Center for Allergy Research and Education (CK-CARE), Davos, Switzerland. [10] Department of Regenerative Medicine and Immune Regulation, Medical University of Bialystok, Bialystok, Poland. [11] Department of Pathophysiology, Jagiellonian University Medical College, Krakow, Poland. [12] Allergy Unit, Department of Dermatology, University Hospital Zürich, Zürich, Switzerland. [13] Pulmonary Division, University Hospital of Zurich, Zurich, Switzerland. [14] National Food Institute, Technical University of Denmark, Copenhagen, Denmark. [15] Depts of Medicine and Microbiology, APC Microbiome Ireland, National University of Ireland, Cork, Ireland. [16] These authors contributed equally: Edith M. Hessel, Liam O'Mahony. *email: liam.omahony@ucc.ie

Changes in lifestyle, diet, weight, environment and microbiome have all been associated with an increased risk and severity of chronic inflammatory disorders such as allergies and asthma[1,2]. However, the complex interactions between these factors and host immunoregulatory processes are still poorly understood at a mechanistic level in humans. In particular, it is unclear if changes in microbiota composition contribute to disease pathology or if these changes reflect altered immune reactivity within host tissues[3]. The microbiota contributes significantly to host health via multiple mechanisms[4,5]. Both the composition and metabolic activity of microbiota have profound effects on proinflammatory activity and the induction of immune effector functions or tolerance within mucosal tissues[6–8]. A diverse microbiota contributes to a balanced homeostatic immunological state, while microbial dysbiosis has been associated with many inflammatory diseases including asthma and obesity[9–11].

Asthma is characterized by recurrent and reversible airflow obstruction with airway inflammation central to its pathogenesis[12]. Several different molecular mechanisms can lead to similar clinical outcomes, which has led to the concept of linking asthma endotypes (an asthma subtype defined by a distinct pathophysiological mechanism) with asthma phenotypes[13]. However, to fully understand the complex interactions between innate and adaptive immune cells in different endotypes, additional factors such as the microbiome, nutrition and host metabolic activity need to be considered so that therapeutic options can be identified and dysregulated mechanisms appropriately targeted.

Since 1980, the number of obese individuals has doubled in more than 70 countries[14]. Correspondingly, the number of obese patients with asthma has also risen dramatically[15]. Obesity has been found to be a distinguishing variable for clustering and classifying asthma subtypes (e.g., enriched in women with adult onset) and the obese asthmatic is more likely to become corticosteroid resistant, has a higher risk of being hospitalized and more frequently presents with severe disease[16–21]. Severe asthma is defined as asthma that requires treatment with high-dose inhaled corticosteroids combined with a second controller and/or systemic corticosteroids to maintain control or, asthma that remains uncontrolled despite this therapy[22]. However, both obese and non-obese asthma patients can present with severe disease and it is currently unknown if similar or divergent factors, such as the microbiome, might contribute to asthma severity independent of obesity[23,24].

In this study, our aim is to comprehensively characterize the immunological and microbiota changes that occur in obese asthma patients and to determine if these changes are related to asthma, to obesity, or both. In addition, we determine if changes in the microbiota associated with asthma severity are shared between obese and non-obese asthma patients. We describe here substantial additive effects of obesity and asthma on host immunological responses and the microbiota. In addition, both obese and non-obese asthma patients with severe disease have reduced fecal levels of *Akkermansia muciniphila*, which may have a causal relationship as suggested by murine models of acute and chronic airway inflammation.

## Results

**Systemic inflammation is enhanced by obesity and asthma**. Patient demographic details and comorbidities are detailed in Table 1 and Supplementary Table 1, respectively. Inflammatory markers were measured in serum of obese asthmatics ($n = 50$), non-obese asthmatics ($n = 52$), obese non-asthmatics ($n = 50$), and their non-obese non-asthmatic healthy counterparts ($n = 47$). The liver acute phase proteins C-reactive protein (CRP),

serum amyloid A (SAA) and fibrinogen were significantly elevated in the serum of obese non-asthma and obese asthma patients (Fig. 1a and Supplementary Fig. 1). Similarly, proinflammatory cytokines and chemokines were significantly elevated in the serum of both obese patient groups. Gene expression analysis of whole blood identified differentially expressed genes (DEGs) in obese asthmatics ($n = 50$), non-obese asthmatics ($n = 53$), obese non-asthmatics ($n = 51$) relative to healthy volunteers ($n = 48$, Supplementary Fig. 2). Significant enrichments in ontologies related to inflammatory and innate immune responses (Fig. 1b) were accentuated in obese asthmatics ($n = 50$), suggesting an additive effect between obesity and asthma.

**Airway inflammation is influenced by obesity and asthma**. We obtained bronchoalveolar lavage (BAL) fluid and bronchial biopsies from obese asthmatics ($n = 10$), non-obese asthmatics ($n = 12$), obese non-asthmatics ($n = 11$) and their non-obese non-asthmatic healthy counterparts ($n = 8$). Acute phase proteins, ICAM-1 and VCAM-1 levels were elevated in BALs from obese patients (Fig. 2a and Supplementary Fig. 3). Non-obese asthma patients ($n = 12$) had the highest levels of chemokines within BAL. However, BAL IL-5 levels were significantly elevated for both obese ($n = 10$) and non-obese asthma ($n = 12$) patients. The total number of inflammatory cells (including eosinophils, neutrophils and lymphocytes) in BAL cytospins was significantly different between the groups ($p = 0.042$, ANOVA). However, while elevated eosinophils, neutrophils, and lymphocytes were observed in specific obese non-asthma, non-obese asthma and obese asthma patients, none of the inflammatory cell types alone were statistically significantly different between the groups (Fig. 2b). The presence of eosinophils and neutrophils was confirmed by H&E staining in available biopsies (Supplementary Fig. 4).

Transcriptomic analysis of bronchial biopsies revealed a number of genes and related pathways that were differentially expressed in obese and asthmatic individuals. The top ten most significant gene ontology pathway enrichments for each group are illustrated in Supplementary Fig. 5a, while expanded heatmaps of immunologically relevant DEGs are illustrated in Supplementary Fig. 5b. Asthma-related gene ontology pathways were enriched in both non-obese asthmatics ($n = 12$) and obese asthmatics ($n = 10$), but not obese non-asthmatics ($n = 11$), compared to non-obese non-asthmatic controls ($n = 8$, Fig. 2c). Both obese groups displayed significant enrichments in pathways relating to airway remodeling and inflammatory responses (Fig. 2c). In BALs, the top ten most significant gene ontology pathway enrichments for each group are illustrated in Supplementary Fig. 6a, while expanded heatmaps of immunologically relevant DEGs are illustrated in Supplementary Fig. 6b. Asthma-related gene ontology pathway enrichments were evident in BALs from non-obese asthmatics, while enrichments in gamma-secretase proteolytic targets, epithelial-to-mesenchymal transition and WNT signaling were observed in both obese groups (Fig. 2d).

**Obesity and asthma influence microbiome composition**. To investigate the relationship between obesity, asthma and the microbiome, we performed 16S rRNA profiling of samples from the lower gastrointestinal tract (gut, $n = 202$), upper gastrointestinal tract (oral, $n = 41$), lower respiratory tract (BAL, $n = 41$) and upper respiratory tract (nasal, $n = 41$). The composition of the microbiota showed significant differences based on body site (Fig. 3a). Specific differences at the genus taxonomic level were observed between groups for the lower respiratory tract (BALs, Fig. 3b), upper respiratory tract (nasal, Fig. 3c), upper gastrointestinal tract (oral, Fig. 3d) and lower gastrointestinal

**Table 1 Patient demographics.**

| | Healthy controls | Non-obese asthma | Obese non-asthma | Obese asthma | p-value |
|---|---|---|---|---|---|
| n | 48 | 53 | 51 | 50 | |
| Age (S.D.) | 39.4 (11.9) | 39.5 (11.3) | 44.4 (13.5) | 44.6 (11.6) | 0.078 |
| BMI (S.D.) | 22.2 (1.4) | 22.9 (1.5) | 35.6 (4.6) | 34.6 (5.1) | <0.001 |
| Male/female | 19/29 | 22/31 | 18/33 | 17/33 | 0.817 |
| Smoker (%) | 18 (38) | 8 (15) | 20 (40) | 9 (18) | 0.011 |
| Alcohol U/week (S.D.) | 2.3 (2.2) | 2.5 (3.5) | 1.8 (2.4) | 1.4 (2.0) | 0.141 |
| FEV1% (S.D.) | – | 81.7 (15.5) | – | 79.8 (17.3) | 0.619 |
| Allergy (%) | – | 40 (75) | – | 36 (72) | 0.823 |
| Daily ICS (%) | – | 38 (72) | – | 40 (80) | 0.365 |
| Daily ICS dose (S.D.)[a] | – | 678 (585) | – | 946 (710) | 0.054 |
| Oral steroids (%) | – | 2 (4) | – | 8 (16) | 0.048 |
| SABA (%) | – | 22 (42) | – | 26 (52) | 0.327 |
| LABA (%) | – | 33 (62) | – | 31 (62) | 1.000 |
| Daily LABA dose (S.D.)[b] | – | 62 (56) | – | 86 (78) | 0.098 |
| Anti-histamines (%) | – | 32 (60) | – | 30 (60) | 1.000 |
| Leukotriene receptor antagonist (%) | – | 27 (51) | – | 27 (54) | 0.844 |
| Daytime asthma symptoms >2 times(%) | – | 10 (19) | – | 18 (36) | 0.076 |
| Exercise/activity limited due to asthma(%) | – | 18 (34) | – | 25 (50) | 0.113 |
| Waking at night due to asthma symptoms(%) | – | 12 (23) | – | 21 (42) | 0.056 |
| Use of rescue medications >2 times(%) | – | 14 (26) | – | 12 (24) | 0.824 |

[a]Dose of inhaled glucocorticosteroids calculated as a budesonide equivalent microgram/day
[b]Dose of LABA calculated as a salmeterol equivalent microgram/day

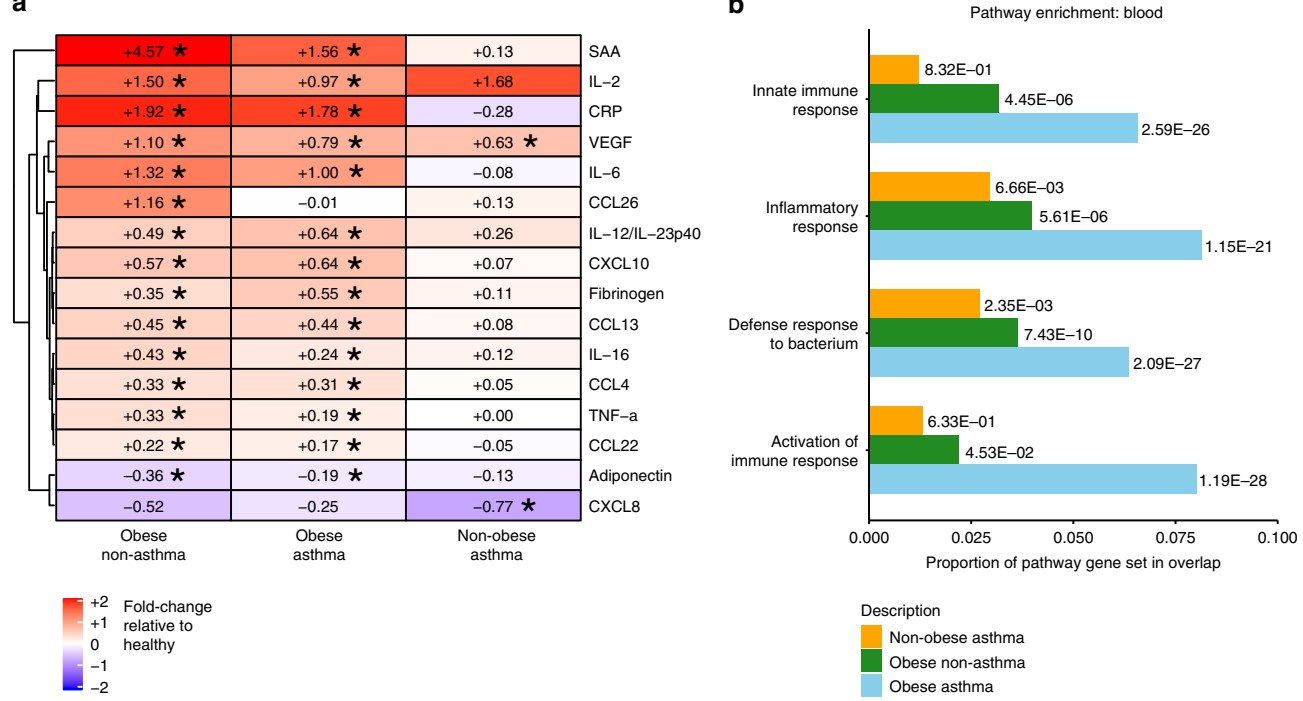

**Fig. 1 Blood inflammatory profiles. a** Fold change of cytokine levels in serum for obese non-asthma ($n = 50$), obese asthma ($n = 50$) and non-obese asthma ($n = 52$) patients relative to the non-obese non-asthmatic group ($n = 47$) are illustrated. All comparisons with a $P < 0.05$ (ANOVA with post hoc Tukey-kramer test) are labeled with asterisks. **b** Gene ontology pathway enrichments in whole blood for obese non-asthma ($n = 51$), obese asthma ($n = 50$) and non-obese asthma ($n = 53$) patients relative to the non-obese non-asthmatic group ($n = 48$) are illustrated using both the proportion of the pathway or ontology gene set that overlapped with the differentially expressed genes in that comparison and the FDR-corrected $p$-values as a label next to the bar. Source data are provided as a Source Data file.

tract (fecal, Fig. 3e), after correcting for known covariates, age and gender (Supplementary Table 2). Interestingly, multiple differences were shared between the obese groups, or the asthma groups, especially within the BAL samples, suggesting that obesity

and asthma have an additive effect on the microbiome alterations associated with asthma in obese individuals. Of note, the relative abundance of *Dehalobacterium* was increased for all asthma patients, independent of BMI. A random-forest classifier model of

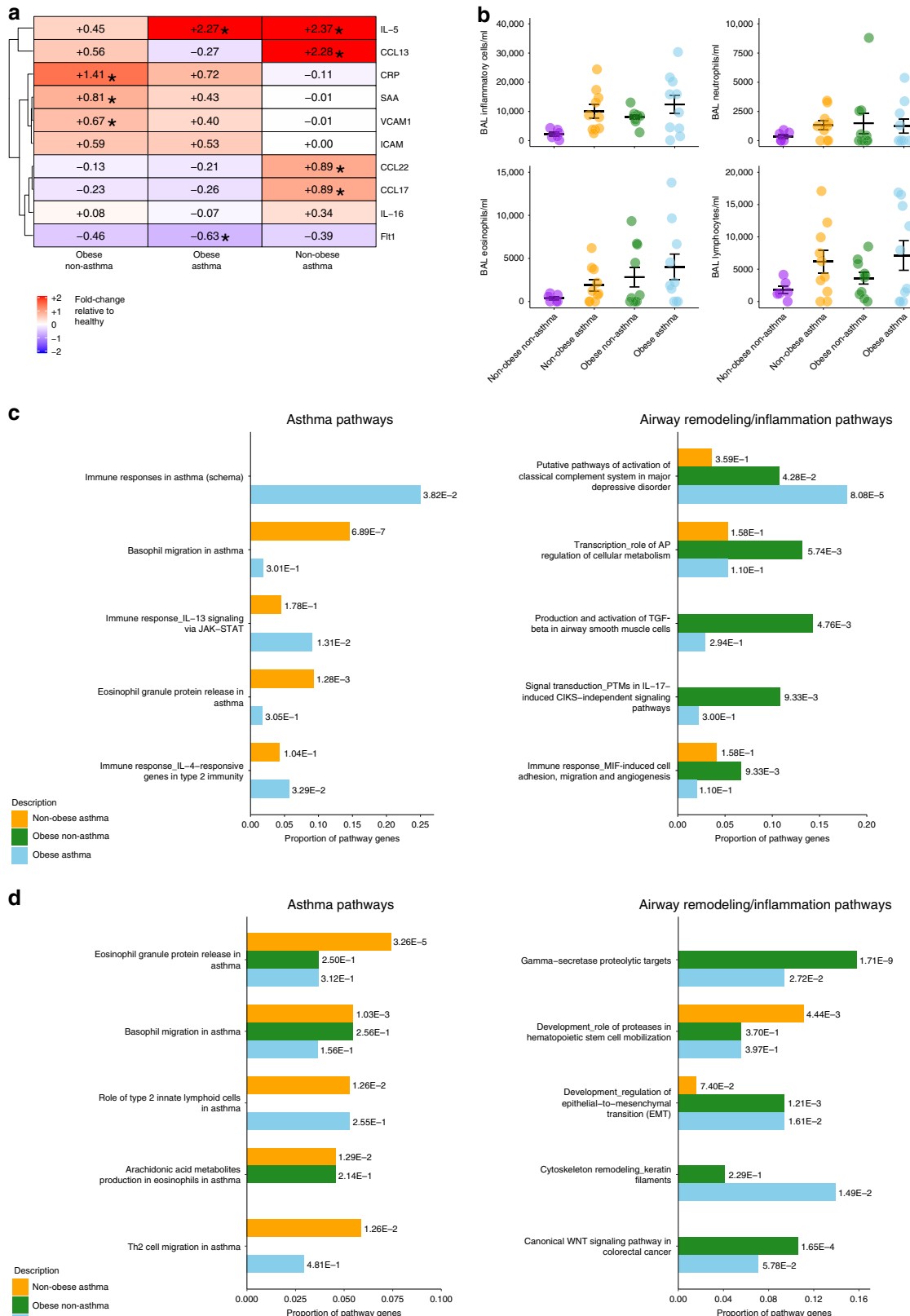

**Fig. 2 Lung inflammatory profiles. a** Fold change of cytokine levels in bronchoalveolar lavages (BALs) for obese non-asthma ($n = 11$), obese asthma ($n = 10$) and non-obese asthma ($n = 12$) patients relative to the non-obese non-asthmatic group ($n = 7$) are illustrated. All comparisons with a $P < 0.05$ (ANOVA with post hoc Tukey-kramer test) are labeled with asterisks. **b** Total inflammatory cell numbers and differential cell counts in BAL cytospins are illustrated (mean $+/-$ standard error). Gene ontology enrichments in asthma-related pathways or airway remodeling/inflammation-related pathways in **c** lung biopsy and **d** BALs for each group (Non-obese asthma ($n = 12$); Obese non-asthma ($n = 11$); Obese asthma ($n = 10$)) compared to non-obese and non-asthma volunteers ($n = 8$), are illustrated using both the proportion of the pathway or ontology gene set that overlapped with the differentially expressed genes in that comparison and the FDR-corrected p-values as a label next to the bar. Source data are provided as a Source Data file.

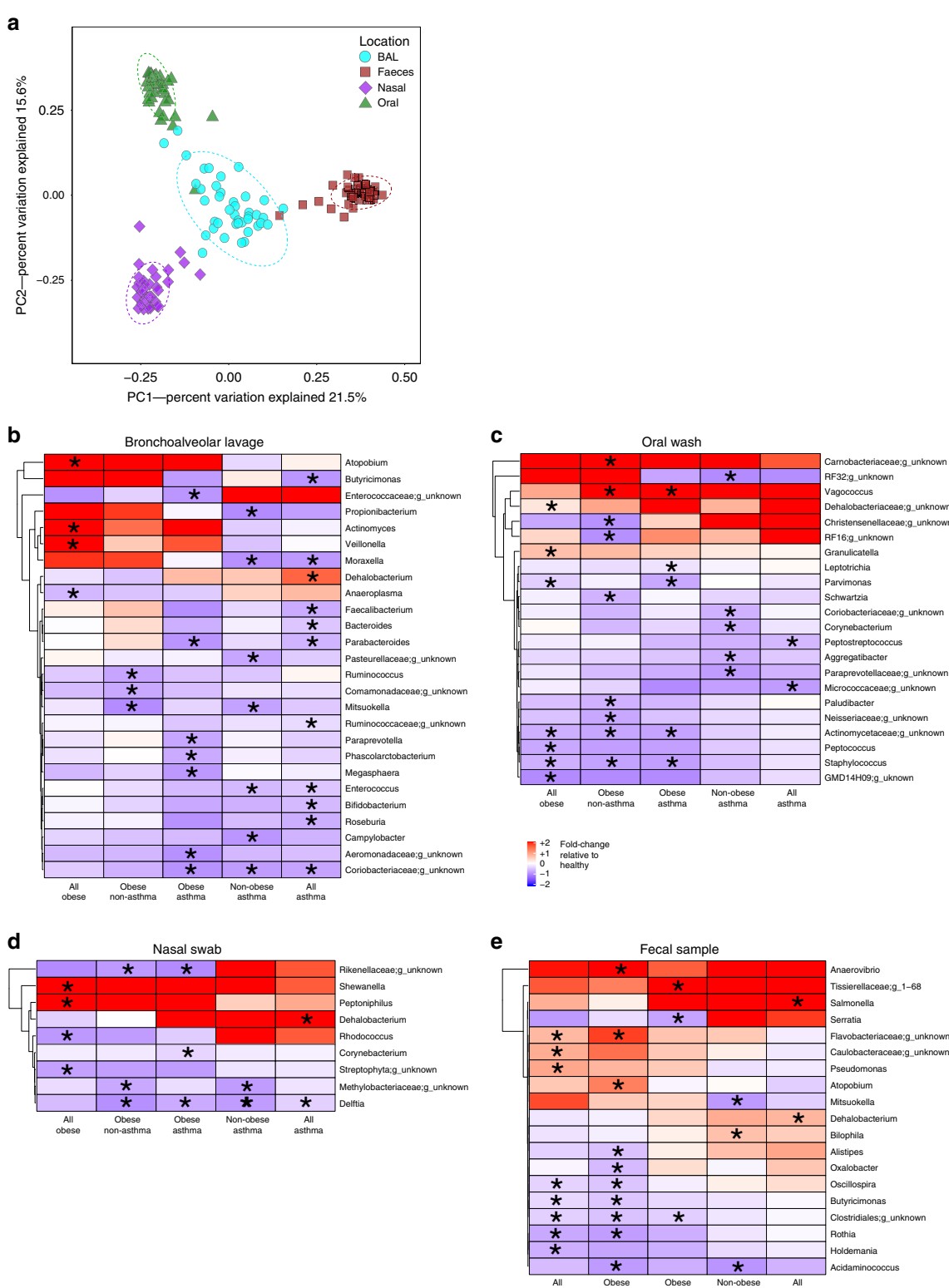

**Fig. 3 Microbiota changes associated with obesity and asthma. a** Distinct body sites show significantly different microbiota compositions (*P* < 0.001, PERMANOVA, Pseudo F-statistic = 41.42), independent of obesity or asthma (*n* = 41, all groups combined). Fold change of genera relative abundances in bronchoalveolar lavages (**b**, *n* = 41), nasal swabs (**c**, *n* = 41) and oral washes (**d**, *n* = 41) for non-obese asthma (*n* = 12), obese non-asthma (*n* = 11), obese asthma (*n* = 10) relative to the non-obese non-asthmatic group (*n* = 8) are illustrated. **e** Fold change of genera relative abundances in fecal samples for non-obese asthma (*n* = 53), obese non-asthma (*n* = 51), obese asthma (*n* = 50) relative to the non-obese non-asthmatic group (*n* = 48) are illustrated. All comparisons with a *P* < 0.05 (ANOVA with post hoc Tukey-kramer test) are labeled with asterisks.

the fecal microbiota had a reasonable ability to distinguish obesity (AUC = 0.76), but little predictive value for asthma (AUC = 0.53). No significant differences were observed for alpha diversity measures between the groups, except for the obese asthmatics who had a significantly increased Shannon diversity within the nose (Supplementary Fig. 7).

**Microbiome correlates with markers of inflammation.** Within the gut microbiota, three distinct bacterial enterotypes were evident, as already described by others[25]. A *Bacteroides*-dominated enterotype (E1, Bacteroides:Prevotella ratio > 2.0), a *Prevotella*-dominated enterotype (E2, Bacteroides: Prevotella ratio < 0.50) and a mixed enterotype (Emix) were observed (Fig. 4a). Patients with a *Prevotella*-rich gut enterotype (E2, $n = 39$) displayed increased serum levels of the chemokines CCL22, CCL13, CCL17, and CXCL10 compared to individuals with an E1 ($n = 116$) or Emix ($n = 37$) enterotype (Fig. 4b). In addition, several serum biomarkers were significantly correlated ($p < 0.01$, Pearson's test) with the levels of specific gut microbes in both obese (Supplementary Fig. 8a) and asthma patients (Supplementary Fig. 8b).

Among BAL samples ($n = 40$), significant correlations were observed between inflammatory cell numbers and microbiome composition (Supplementary Table 3). Of note, increased eosinophils in asthma patients, regardless of their BMI, was associated with an increased relative abundance of the genera *Rothia*, *Dorea*, *Lautropia*, and *Haemophilus*. We explored two major variables, smoking status and inhaled corticosteroids, for their effect on oral, nasal, and BAL microbiome, and observed no statistically significant differences between these groups in the microbiome composition at each airway site (PERMANOVA $p > 0.05$ for each). In contrast, significant associations ($p < 0.01$, Pearson's test) between BAL cytokine levels with individual microbial genera were observed in obese patients (Supplementary Fig. 9a) and asthma patients (Supplementary Fig. 9b). In addition, focusing only on the taxa that were shown to be significantly decreased within the asthma BALs (Fig. 3b), their combined relative abundance inversely correlated with BAL IL-5, IFN-γ, and IL-15 levels ($n = 40$, Fig. 4c).

***Akkermansia muciniphila* decreases in severe asthma patients.** In addition to the effects of obesity on the gut microbiota of asthma patients, we evaluated if asthma disease severity could also influence the composition of the gut microbiota. In patients with severe asthma ($n = 41$), a significant reduction in the family *Verrucomicrobiaceae* was observed, compared to patients with mild/moderate asthma ($n = 53$, Fig. 5a). The most common species within this family is *Akkermansia muciniphila*, which showed a significant decrease ($p = 0.0041$) in its relative abundance in severe asthma patients (Supplementary Fig. 10). The reduced level of *A. muciniphila* was specifically associated with severe asthma as the relative abundance of *A. muciniphila* was not significantly different between the asthma or obese groups (non-obese non-asthmatic 0.046 +/− 0.082, non-obese asthmatics 0.035 +/− 0.069, obese non-asthmatics 0.026 +/− 0.028, obese asthmatics 0.046 +/− 0.076, mean plus/minus (+/−) standard deviation, $p > 0.05$ ANOVA). Both obese and non-obese asthmatics with severe disease ($n = 22$ and $n = 19$, respectively) showed a significant decrease in *A. muciniphila* compared to obese and non-obese patients with mild/moderate asthma ($n = 23$ and $n = 30$, respectively, Fig. 5b). Using qPCR primers specific to *A. muciniphila*, we observed a strong concordance with the relative abundance data (rho = 0.762) and we confirmed the reduced levels in *Akkermansia* associated with severe disease (Fig. 5c). In asthma patients, we also observed a negative correlation between *A. muciniphila* and circulating CRP levels

(Fig. 5d). These data suggest that lower levels of *A. muciniphila* are associated with a higher risk of severe asthma symptoms.

***A. muciniphila* reduces airway inflammation in animal models.** In order to determine if any of the microbial changes described above might play a causal role in influencing respiratory inflammation, we selected *A. muciniphila* to test further in murine models of allergen-induced respiratory airway disease. Oral administration of *A. muciniphila* induced a marked reduction in BAL eosinophil numbers in female animals sensitized and challenged with ovalbumin (OVA), compared to animals that received OVA alone ($n = 5$ mice per group, Fig. 6c). In addition, IL-4 and IL-5 secretion from OVA-stimulated isolated lung cells was reduced in animals treated with *A. muciniphila* (Fig. 6c). Heat killed *A. muciniphila* or cell free supernatants from *A. muciniphila* cultures did not reduce OVA-induced eosinophilia or cytokine levels (Fig. 6c). *A. muciniphila* administration, but not heat killed *A. muciniphila* or its supernatant, was associated with an altered lymphocyte profile within lung tissue as the percentage of IL-4 and IFN-γ positive CD4 T cells were reduced, while IL-10 + Foxp3 + double positive lymphocytes were increased ($n = 5$ mice per group, Fig. 6d and Supplementary Fig. 11). *A. muciniphila* was equally effective in reducing BAL inflammatory cell numbers in male mice ($n = 6$ per group, Fig. 6e). Airway hyper-reactivity in response to methacholine was significantly reduced in animals administered *A. muciniphila* ($n = 8–9$ mice per group, Fig. 6f). Oral administration of *A. muciniphila* significantly reduced the number of BAL inflammatory cells in the acute house dust mite (HDM) extract challenge model ($n = 7–8$ mice per group, Fig. 6g). *A. muciniphila* was equally effective in reducing BAL inflammatory cell numbers in MyD88$^{−/−}$ animals ($n = 5$ mice per group, Fig. 6h). *A. muciniphila* levels in fecal samples increased 1000–10,000 fold in exposed mice, while *A. muciniphila* was not detected in the BALs from exposed animals ($n = 3$ mice per group at each time point, Fig. 6i).

The influence of *A. muciniphila* on airway inflammation was also assessed in a chronic model of HDM exposure, in which persistent airway inflammation was initiated prior to administration of *A. muciniphila*. Administration of *A. muciniphila* reduced the number of all innate and adaptive cell types examined within the BAL, suggesting that *A. muciniphila* accelerated the resolution of airway inflammation following cessation of HDM exposure ($n = 6$ mice per group, Fig. 7b and Supplementary Fig. 12). In additional groups of animals, mice chronically exposed to HDM were allowed to recover in the absence of HDM and were then re-challenged with a single high dose of HDM extract. A significant influx of eosinophils was observed in the BAL within 24 h post re-challenge ($n = 6–8$ mice per group at each time point, Fig. 7d). However, re-challenged animals that received *A. muciniphila* during the resolution phase displayed a significantly reduced eosinophil response (Fig. 7d). Of note was the reduction in infiltration of a sub-set of eosinophils expressing high levels of Sialic acid-binding immunoglobulin-type lectin-F (Siglec$^{−}$F$^{hi}$) ($n = 6$ mice per group, Fig. 7e). Representative dot-plots illustrating the presence of Siglec$^{−}$F$^{hi}$ eosinophils in the lung are illustrated in Fig. 7f.

## Discussion
In this study we describe immunological and microbiome alterations that are associated with obesity and asthma. In addition, we clearly identify an obese asthma phenotype that shares immunological and microbiome features of both obesity and asthma. Furthermore, we discovered that asthma severity was associated with reduced levels of *A. muciniphila* in the gut, which may be clinically relevant as this bacterium was protective in

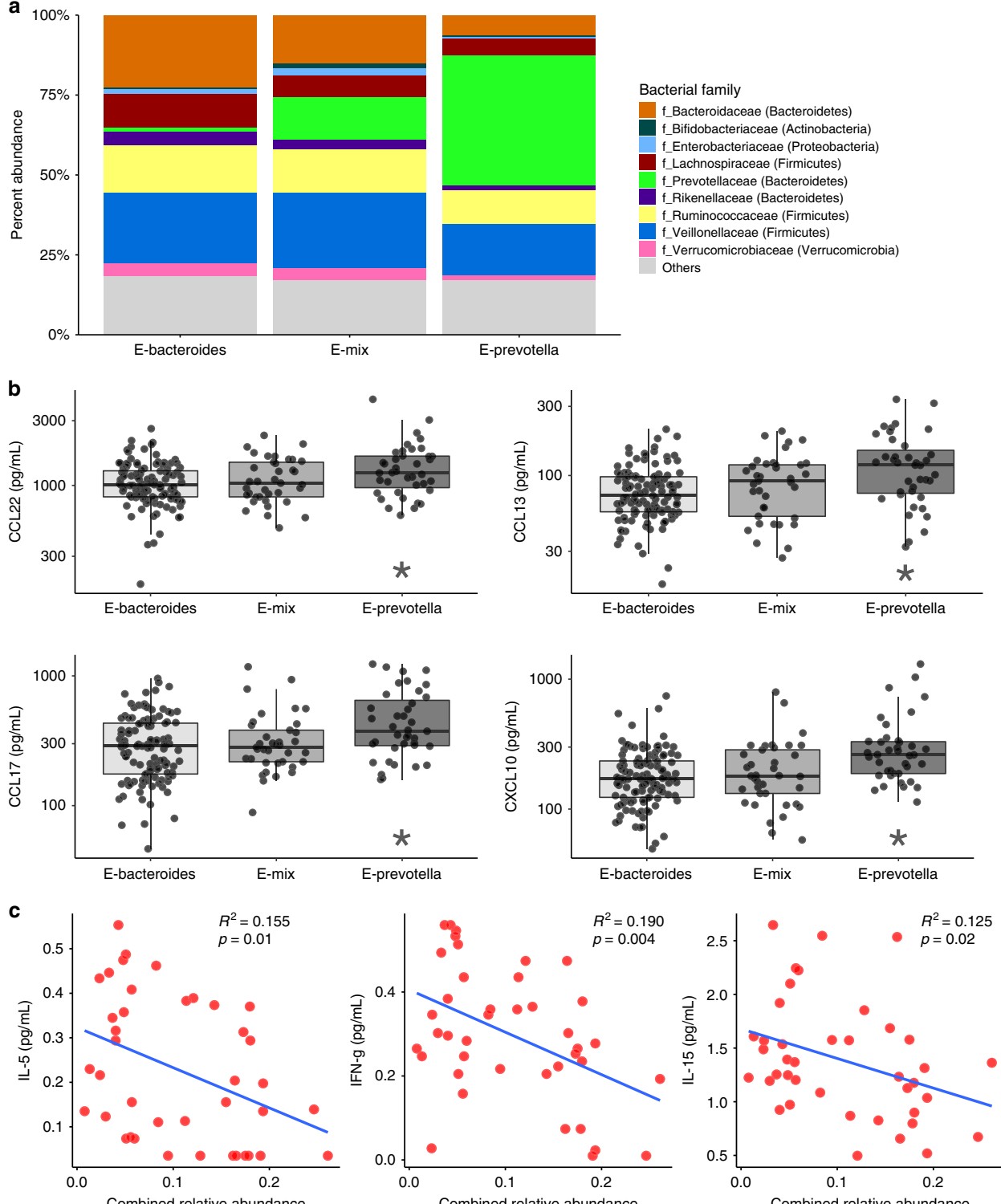

**Fig. 4 Microbial community composition correlates with cytokine levels. a** A *Bacteroides*-dominated enterotype, a *Prevotella*-dominated enterotype and a mixed enterotype were observed in 60%, 21%, and 19%, respectively of volunteer fecal samples (*n* = 202). Average abundances of the other dominant bacterial families (not *Bacteroides* or *Prevotella*) did not differ significantly between enterotypes. **b** Serum CCL22, CCL13, CCL17, and CXCL10 are significantly higher (ANOVA and Tukey correction) for individuals with a *Prevotella*-dominated gut microbiota (E2, *n* = 39), compared to individuals with a *Bacteroides*-dominated gut microbiota (E1, *n* = 116) or a mixed microbiota (Emix, *n* = 37). Box plots show median and whiskers represent 10–90 percentiles. **c** The combined relative abundance for taxa reduced in the asthmatic lung (*Butyricimonas, Moraxella, Propionibacterium, Pasteurellaceae, Campylobacter, Faecalibacterium, Bacteroides, Parabacteroides, Mitsuokella, Megasphaera, Ruminococcaceae, Paraprevotella, Phascolarctobacterium, Roseburia, Enterococcus, Bifidobacterium, Aeromonadaceae,* and *Coriobacteriaceae*) negatively correlated with BAL IL-5, IFN-γ, and IL-15 levels (*n* = 40, all groups combined) using linear regression analysis. Source data are provided as a Source Data file.

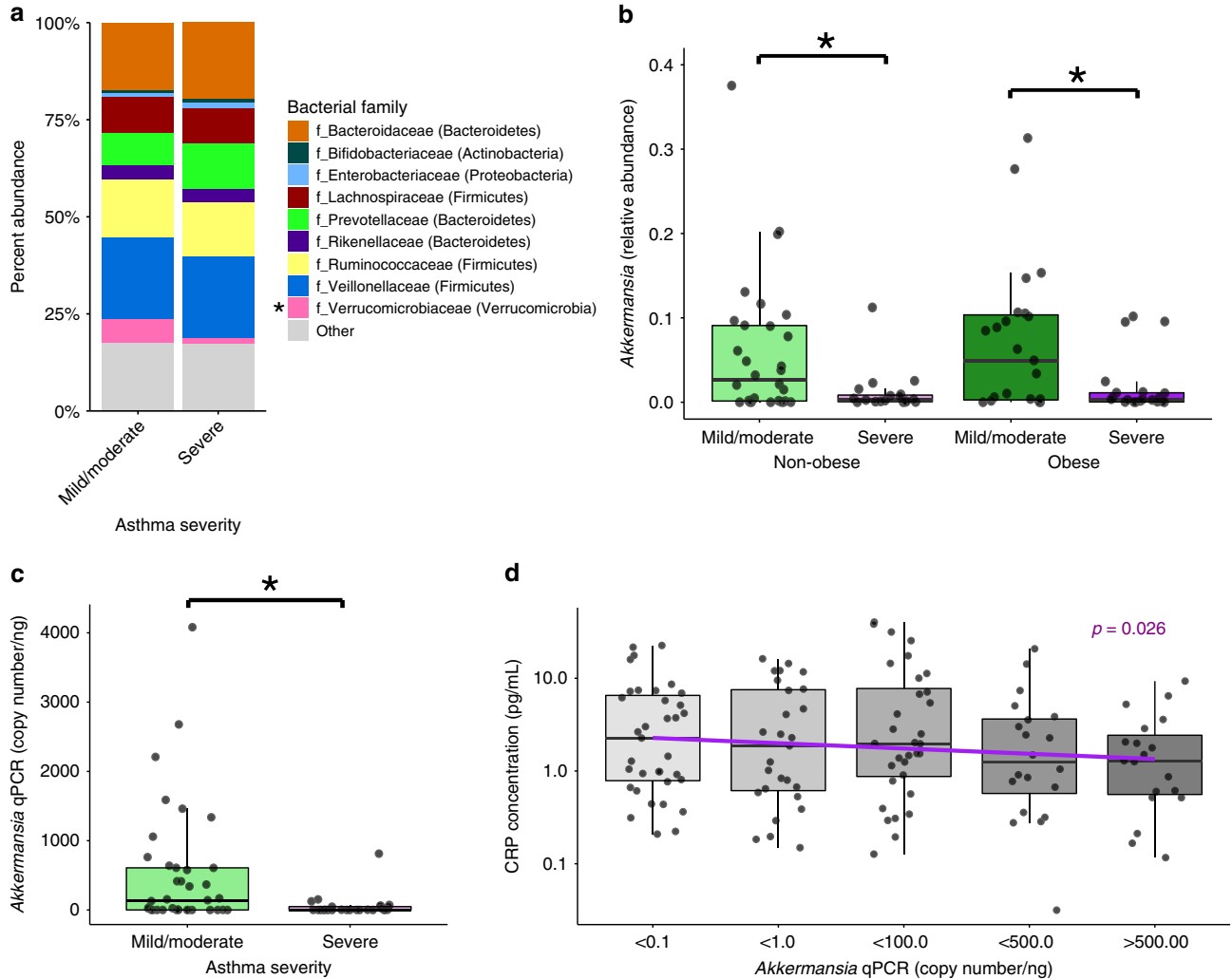

Fig. 5 **A. muciniphila is negatively correlated with severe asthma. a** Relative abundances at the Family taxonomic level within the fecal samples of asthma patients with mild/moderate ($n = 53$) or severe ($n = 41$) disease are illustrated Asterisk (*) denotes a $P < 0.05$ ANOVA with post hoc Tukey-kramer test). **b** Relative abundance of the genus *Akkermansia* is significantly decreased in non-obese severe ($n = 19$) compared to non-obese mild/moderate ($n = 30$) disease and obese severe ($n = 22$) compared to obese mild/moderate ($n = 23$) disease (Mann-Whitney test). **c** Absolute qPCR quantification of *Akkermansia muciniphila* in severe ($n = 28$) versus mild/moderate ($n = 38$) disease (Mann-Whitney test). **d** Decreased levels of *Akkermansia muciniphila* is associated (ANOVA with post hoc Tukey-kramer test) with increased levels of C-reactive protein (CRP). Box plots show median and whiskers represent 10–90 percentiles. Source data are provided as a Source Data file.

multiple murine models of both acute and chronic respiratory airway disease.

The combination of obesity and asthma had significant effects on host inflammatory and transcriptomic responses. This was clearly observed in the peripheral blood transcriptome, where enrichments in ontologies related to inflammatory and innate immune responses were accentuated in obese asthmatics, suggesting an additive effect between obesity and asthma. Within BALs and lung biopsies, non-obese asthma patients showed enrichments for TH2 and asthma-related ontologies, while obese non-asthma patients showed enrichments for tissue-remodeling-related and inflammation-related ontologies. Obese asthmatics displayed enrichments in tissue remodeling-related, inflammation-related and TH2-related ontologies. However, the quantitative additive effect observed for specific ontology enrichments in peripheral blood was not observed for BALs or lung biopsies, rather the obese asthmatic displayed qualitative additive effects in gene ontologies representing both asthma and obesity signatures within the lung. This difference between systemic versus pulmonary-specific responses

may reflect the contribution of multiple organs (e.g., liver and adipose tissue) to systemic inflammatory gene responses[26].

Previous studies have demonstrated that obesity is associated with an altered gut microbiota[25,27,28]. Our data extends these findings in showing that the composition of the microbiota changes in both the obese and asthma states relative to healthy individuals, not only in gut and lung, but also in the nose and mouth. This suggests that the microbiota can be altered at sites distant to the diseased organ, perhaps due to the influence of diet, inflammation or medications. Certain microbiota changes in the obese asthmatic were shared by obese individuals or by patients with non-obese asthma. Thus, the additive effect of having asthma and being obese exaggerates the microbial changes present in these patients. In obese asthmatics, atopic status is likely to also play a role in better phenotyping this patient group, particularly their response to treatments such as weight loss. However, the majority of obese asthmatics described in this cohort are atopic and therefore it was not possible to further separate microbiome-host effects associated with atopic status in obese asthmatics. One additional finding is the identification of

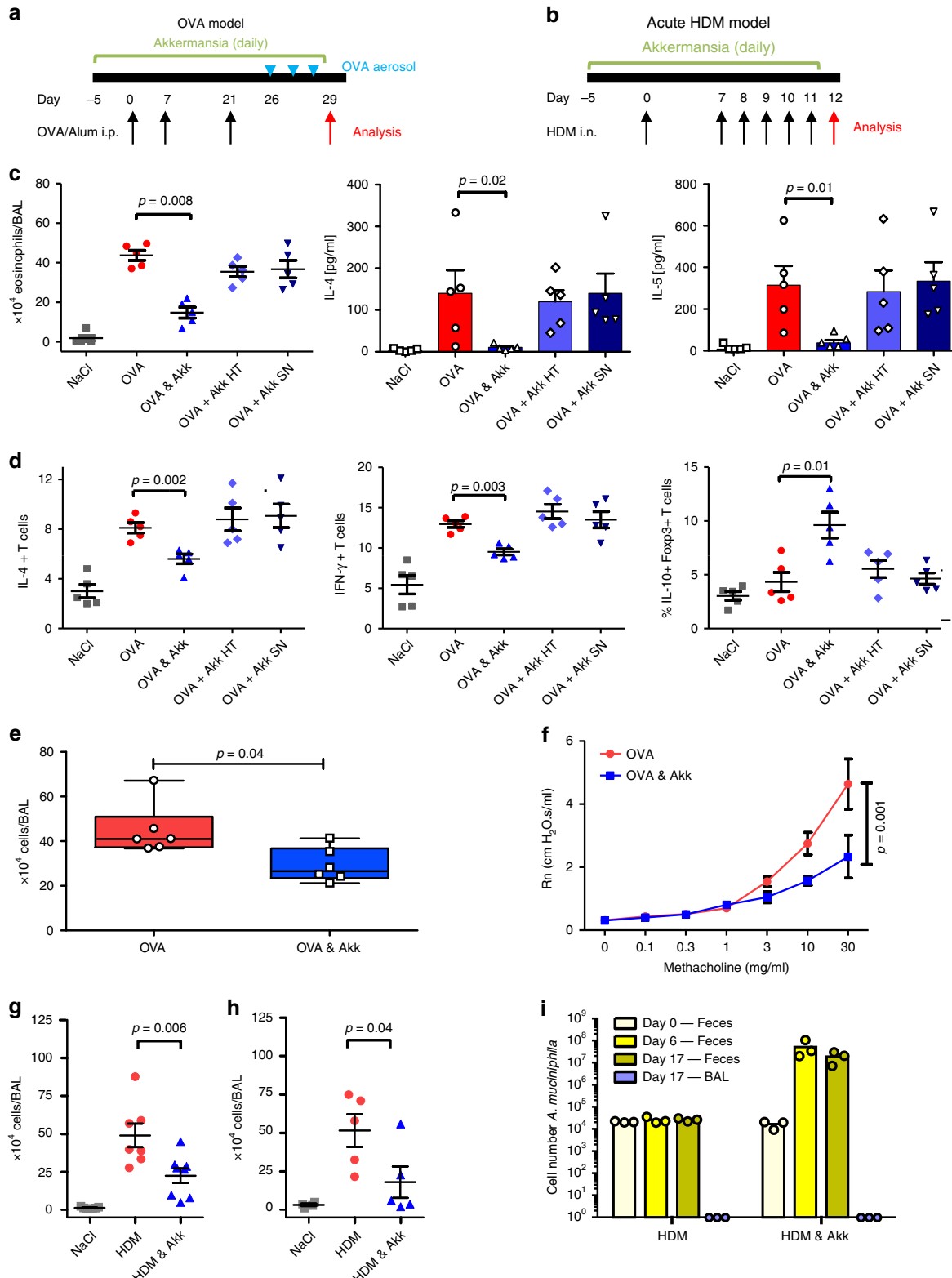

*Dehalobacterium* in obese and non-obese asthma patients. *Dehalobacterium* is a strictly anaerobic dichloromethane-degrading bacterium[29,30]. Dichloromethane present in the environment is the result of industrial emissions and increased levels of *Dehalobacterium* may therefore indirectly indicate greater exposure to solvents containing this compound. However, further work is required to determine if an increase in this organism has any influence on host immune responses.

Significant associations were observed between the composition of the microbiota and host inflammatory responses, reinforcing the hypothesis that there is a close relationship between immune regulatory mechanisms and the microbes that reside within us. For example, the highest serum chemokine levels were observed in those with a *Prevotella*-dominated gut microbiota. This association was observed regardless of obesity or asthma status, suggesting a causal relationship. In addition,

**Fig. 6 A. muciniphila protects against airway inflammation in acute murine models.** The timings for allergen and bacterial administration in the **a** acute ovalbumin (OVA) and **b** acute house dust mite (HDM) models are illustrated. **c** Treatment with viable A. muciniphila cells (OVA + Akk), but not heat killed (Akk HT) A. muciniphila or A. muciniphila filtered culture supernatant (Akk SN) reduces BAL eosinophil numbers and ex vivo allergen stimulated IL-4 and IL-5 secretion in the ovalbumin (OVA) model (n = 5 mice per group). **d** The proportion of CD3$^+$CD4$^+$ T cells from lung tissue homogenates that were IL-4$^+$ or IFN-γ$^+$ were reduced, while the IL-10$^+$Foxp3$^+$ T cells increased, only in animals exposed to viable A. muciniphila (n = 5 mice per group). **e** A reduction in BAL inflammatory cells was also observed for male mice (n = 6 mice per group). The box plot shows median and whiskers represent 10–90 percentiles. **f** Airway hyper-responsiveness induced by increasing doses of methacholine were significantly less (2way ANOVA) in mice administered A. muciniphila by oral gavage (n = 8), compared to control animals administered a placebo gavage (n = 9). Oral administration of A. muciniphila significantly reduced the number of BAL inflammatory cells in the acute house dust mite (HDM) challenge model in **g** wild-type (n = 7 mice per group) and **h** MyD88$^{-/-}$ animals (n = 5 mice per group). **i** A. muciniphila fecal, but not BAL, levels substantially increase in mice administered the bacterium, as measured by qPCR (n = 3 per group per time point). Cell number was estimated using a standard curve of known numbers of bacteria. Unless otherwise indicated, statistical significance for two group comparisons were estimated using Mann-Whitney tests, while mean +/− SE are illustrated in each dot-plot and bar charts. Source data are provided as a Source Data file.

the close relationship between BAL microbiota composition and GM-CSF levels in asthma patients is intriguing as GM-CSF has been shown in murine models to influence lung TH2-associated sensitization[31].

A. muciniphila is a mucin-degrading organism and has been previously associated with protective effects in obesity models[32,33]. However, we found that A. muciniphila was reduced in asthma patients with severe disease, regardless of their BMI. In addition, the experimental models were performed using lean mice, suggesting an influence on lung inflammatory responses independent of its effects on obesity-associated inflammation. While all inflammatory cells were reduced in the murine inflamed lungs, a surprising finding was the significant reduction in eosinophils expressing high levels of Siglec F. These eosinophils are IL-5-dependent and have been shown to facilitate skewing towards a TH2 response[34]. Thus, the inhibitory effect of A. muciniphila on this eosinophil population may have clinical relevance given the established relationship between IL-5 levels, eosinophils and asthma exacerbations[35]. The A. muciniphila mechanism of action is not MyD88-dependent, but may involve other pattern recognition receptors that are MyD88 independent or previously described mechanisms such as improvement of gut barrier integrity or release of metabolites such as nicotinamide that can have systemic effects[36,37]. The increase in IL-10$^+$Foxp3$^+$ lymphocytes may also mediate anti-inflammatory effects within the lung and A. muciniphila administration has previously been shown to increase Tregs in other murine models[38]. In addition, heat killed A. muciniphila was not effective suggesting that either heat sensitive factors or viable bacteria are required for this protective effect and any metabolites secreted in vitro were not sufficient to reduce airway inflammation, at least for the concentrations tested.

An interesting finding of our study is that obese non-asthmatic patients show inflammatory changes within the lung. In addition, it is not only the gut microbiota, but also the microbiota of the mouth, nose, and lung that are altered in obese non-asthmatics. Obese individuals have a higher risk of developing asthma compared to lean individuals[15] and these immune and microbiome changes may play a part in the increased susceptibility of obese individuals in the development of asthma. Indeed, some of these changes may represent pre-asthma features[39]. For example, elevated BAL ICAM-1 and VCAM-1 levels in obese individuals are suggestive of significant vascular injury within the lung, while the increased level of Moraxella may be clinically relevant and should be investigated further[40].

In conclusion, the microbiome and host immune responses are intimately connected at multiple body sites and can be influenced by obesity, asthma and asthma disease severity. The presence of two diseases, obesity and asthma, are additive and contribute to exaggerated inflammatory and microbiotic changes, suggesting

that there's a need to address obesity during asthma management. While the obese asthma patient obviously needs to be treated differently than a non-obese asthma patient, severe asthma may benefit from similar microbiota interventions (i.e., A. muciniphila), regardless of BMI. However, many of the other changes in microbiome composition that we identified in this study may also be biologically relevant and will need to be investigated in future studies for their relative contributions to the stratification and selection of patients for specific therapies.

## Methods

**Patient groups.** A total of 202 volunteers were recruited for this study—obese asthma (n = 50), obese non-asthma (n = 51), non-obese asthma (n = 53), and non-obese non-asthma healthy controls (n = 48). Asthma patients had a physician diagnosis of asthma. Severe asthma was defined according to the American Thoracic Society (ATS) Workshop on Refractory Asthma 2000 report and by the 2013 European Respiratory Society (ERS)/ATS guidelines[22,41]. Obesity was defined as having a body mass index (BMI) greater than 30 kg/m$^2$. Non-obese individuals had a BMI of 20–25 kg/m$^2$. All relevant ethical regulations for work with human participants were complied with, and informed consent was obtained from all participants. The human biological samples were sourced ethically and their research use was in accordance with the terms of the informed consents. Patients were recruited under informed consent at two centers, ALL-MED Medical Research Institute, Wroclaw, Poland and the Pulmonary Division, University Hospital of Zurich, Switzerland. Ethical approval was granted at both sites from the local ethical committee for all study procedures (KEK-ZH-Nr. 2012–0443—Kantonale Ethik-Kommission Zürich; KB-70/2013 and KB-567/2014—Bioethical Committee, Wroclaw Medical University). Patient demographic details and comorbidities are detailed in Table 1 and Supplementary Table 1, respectively.

**Measurement of inflammatory mediators and cells.** Serum was obtained by allowing blood collection tubes (with no anti-coagulant) to rest at room temperature for 1 h and then tubes were centrifuged at 800×g for 10 min. Serum was removed, aliquoted in 500 µl quantities and stored at −80 °C for later analysis. BAL was obtained and filtered through a 70 µm filter into sterile tubes. Aliquots of 500 µl were stored at −80 °C for later analysis. BAL cells were centrifuged onto slides and following air drying were stained using the Diff-Quik stain or remained unstained. Fixed slides were stored at −20 °C and differential cell counts were performed by two independent histopathologists. All soluble mediators were measured using the mesoscale discovery platform (MSD) kits according to manufacturer's instructions. Data were analyzed using one-way ANOVA (correcting for gender) and Tukey's correction.

**Gene expression analysis.** Peripheral blood was collected in Paxgene tubes and immediately frozen at −80 °C until later analysis. Total mRNA was extracted from peripheral blood using Qiagen RNeasy kit (Qiagen, Valencia, CA) and quantified by NanoDrop (Thermo Fisher Scientific, Waltham, MA). RNA integrity was confirmed using the Agilent 2100 BioAnalyzer (Agilent, Palo Alto, CA). Samples were normalized to the lowest concentration sample, and cDNA was made using Superscript Vilo cDNA synthesis Master Mix (Invitrogen Life Technologies, Grand Island, NY). The cDNA samples were then labeled with biotin with the FL-Ovation cDNA Biotin Module V2 and hybridized to a Human Genome U133 Plus 2.0 Array using a Genechip Hybridization Kit. The microarray chips were washed and stained using a Genechip Hybridization Wash and Stain Kit and then scanned using a Genechip Scanner. All reagents and readers were used according to the manufacturer's instructions (Affymetrix, Santa Clara, CA).

The microarray gene expression data were analyzed using ArrayStudio 7.0 (OmicSoft, Cary, NC). Data from.CEL files from Human Genome U133 Plus 2.0

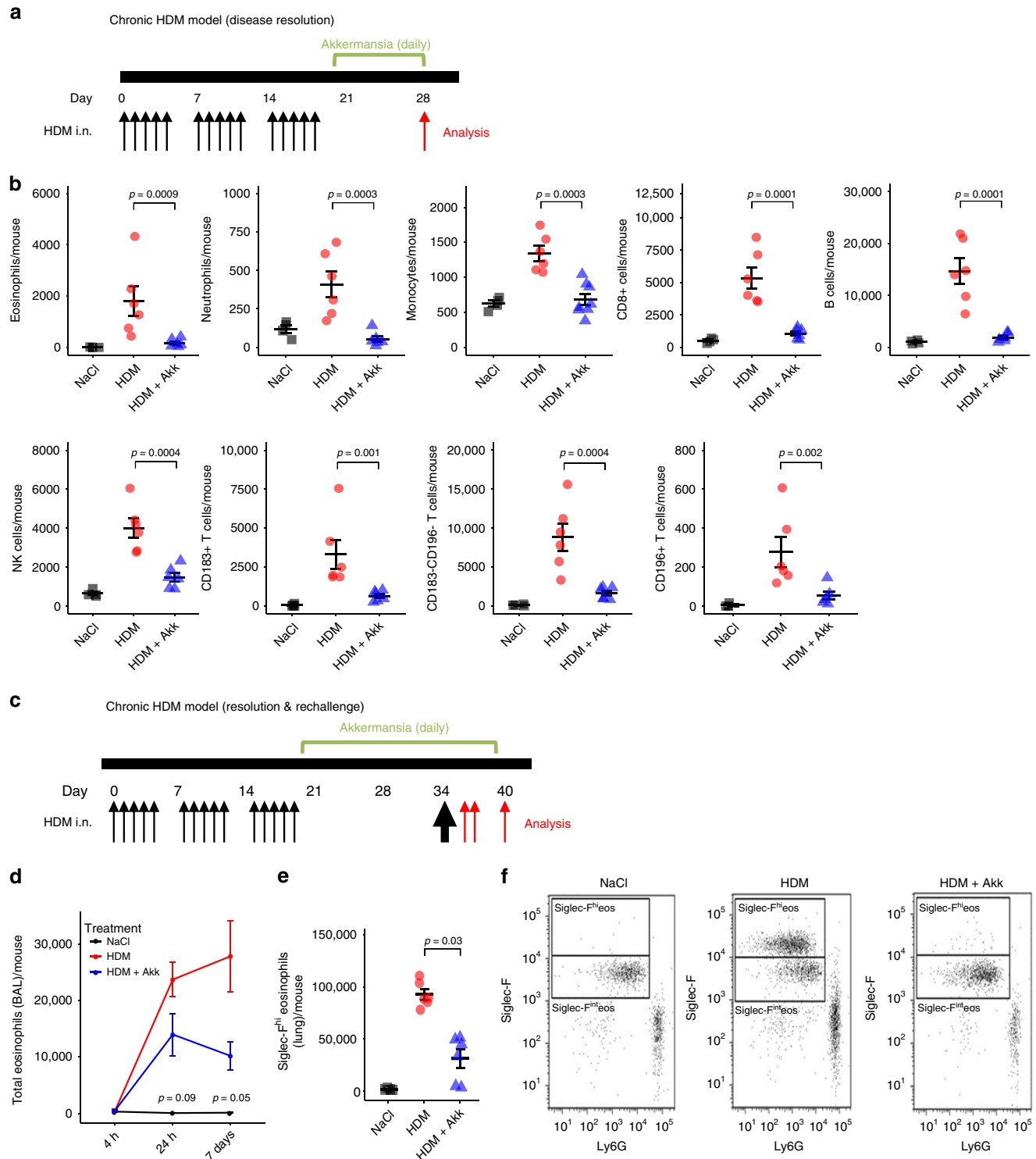

**Fig. 7 *A. muciniphila* protects against airway inflammation in a chronic HDM murine model. a** The timings for allergen and bacterial administration in the chronic HDM model assessing effects on disease resolution are illustrated. **b** Administration of *A. muciniphila* (HDM + Akk) reduced the number of eosinophils (CD45+CD11c−CD11b+MCHII^lo^CD24+Siglec−F+), neutrophils (CD45+CD11c−CD11b+MHCII^lo^CD24+Ly6-G+), monocytes (CD45+CD11c−CD11b+CD24^lo^MHCII^lo^CD64^lo^Ly6-C+), CD8 T cells (CD45+TCRβ+CD8+), B cells (CD45+CD19+), NK cells (CD45+CD49b+), and CD4 T cell subsets (CD45+TCRβ+CD4+CD44+CD62L−CD183+/−CD196+/−) following chronic exposure to HDM (*n* = 6 per group). **c** The timings for allergen and bacterial administration in the chronic HDM model assessing effects on high dose allergen challenge following disease resolution are illustrated. **d** HDM re-challenged animals that received *A. muciniphila* during the resolution phase displayed a reduced eosinophil response at 24 h (*n* = 5 mice HDM group, *n* = 8 mice HDM + Akk group) and 7 days (*n* = 6 mice HDM group, *n* = 6 mice HDM + Akk group) following HDM rechallenge, **e** especially for the Sialic acid-binding immunoglobulin-type lectin-F high (Siglec-F^hi^) eosinophils (*n* = 6 per group). **f** Representative dot-plots illustrating the presence of Siglec-F^hi^ eosinophils in the lung. Statistical significance for two group comparisons were estimated using Mann-Whitney tests and results are shown as mean +/− SE. Source data are provided as a Source Data file.

Affymetrix chips were normalized using robust multiarray averaging and scaled to a mean target intensity of 150.

BAL cells and bronchial biopsies were immediately placed in RNAlater and stored frozen until RNA extraction was performed. Total RNA from BAL cells and bronchial biopsies was extracted using the RNeasy Mini kit (Qiagen) and RNA integrity was confirmed using the Agilent 2100 BioAnalyzer (Agilent, Palo Alto, CA). Samples were required to have a RIN of at least 7.0. Sequencing libraries were prepared from RNA from each sample using the Illumina TruSeq Stranded RNA HT sample preparation kit with RiboZeroTMGold. Libraries were barcoded and then sequenced using an Illumina HiSeq 2000 with 51 bp paired-end reads for targeted coverage of 50 million paired reads per sample. Quality control and trimming of the unaligned FASTQ files was performed using the FASTX-Toolkit (version 0.0.14). Read pairs for each biological replicate were aligned to the human genome version hg38 using STAR (version 2.3.0e_r291) with a STAR index using the default parameters. Samtools (version 0.1.18) was used to convert the aligned SAM files to BAM files[42,43]. Read counts were determined using HTSeq (version 0.6.0) using the htseq-count command with the default parameters (–m union –r pos –s reverse) to define gene-levels counts from the Ensembl v85 annotated.gtf file[44].

**Microbiota analysis**. Fecal samples were collected by volunteers at their home, immediately chilled using ice-packs and delivered to the lab within 4 h. Upon receipt in the lab, fecal samples were separated into 1 g aliquots and stored at −80 °C until analysis. Pre-procedure bronchoscope wash fluid (10 ml), BAL (10 ml), and mouth wash fluid (2 ml) were centrifuged at high speed (22,500×$g$) for 30 min (Hermle Z 231 M microcentrifuge; Hermle Labortechnik GmbH, Wehingen, Germany) in dolphin-nosed eppendorf tubes and the pellets stored at −80 °C until the time of DNA extraction. Nasal swabs of the middle meatus were stored in preservative (Norgen Biotek, Thorold, ON) and immediately frozen. DNA was isolated using PSP Spin Stool DNA Plus Kit (Cat#10381102, Invitek, Berlin, Germany) according to the manufacturer's instructions. Each DNA sample was quantified by spectrophotometry. PCR amplification of the 16S rRNA V4 region was conducted with primers, 515f (5'-GTGCCAGCMGCCGCGGTAA-3') and 806r (5'-GGACTACHVGGGTWTCTAAT-3') including the Illumina sequencing adapters and a 12 bp error-correcting Golay barcode sequence[45,46]. Each 50 µl PCR reaction contained 100 ng of genomic DNA, 2× Phusion High-Fidelity PCR Master Mix with HF Buffer (Cat# M0531L, New England Biolabs, Inc., Ipswich, MA), and 0.2 µM of each primer (Integrated DNA Technologies, Coralville, IA). PCR was performed on an ABI 9700 thermocycler and included the following cycling steps: Initial denaturing at 98 °C for 5 min followed by 40 cycles of 98 °C ×30 s, 60 °C ×45 s, and 72 °C ×1 min ending with a 72 °C 1× min extension. PCR products from the extracted DNA sample were run on a 2.0% TAE agarose gel, excised and purified using QIAquick Gel Extraction Kit (Cat# 28704, Qiagen, Valencia, CA). PCR products were quantitated using Quant-iT PicoGreen dsDNA reagent (Cat # P7589, Invitrogen, Eugene, OR).

To check for proper cluster density and sample normalization, an Illumina MiSeq single-end 26 bp + 12 bp index sequencing run was performed using the MiSeq instrument. The pool was mixed with a PhiX library (Illumina, San Diego CA) at a ratio of 1:9 in order to increase the entropy of the library. A final MiSeq 2 × 150 bp + 12 bp index sequencing run was performed on the pooled samples. Reads were first filtered to remove the PhiX library reads. All reads mapping to the Enterobacteria phage PhiX 174 reference genome (GenBank: NC_001422.1) using the software Bowtie (v1.0.1) were removed from the analysis[47]. The paired reads were next merged with the software PEAR (v0.9.5)[48]. Successfully assembled reads were analyzed using the QIIME software package (v1.8) with the default quality control parameters[49]. Chimeric sequences were identified and removed from the dataset using the UCHIME (v6.1) method[50]. The closed-reference QIIME protocol was used with the UCLUST method to select operational taxonomic units (OTUs)[51]. Sequences with at least 97% identity were clustered together. A representative sequence from each cluster was used to identify bacterial taxa from the May 2013 edition of the Greengenes 16S rRNA database (13_8)[52,53].

In order to confirm the 16s sequencing data, *A. muciniphila* levels in fecal samples were also quantified using PCR. Quantitative PCR analysis was performed using an Applied Biosystems 7900 HT Fast Real-Time PCR system and the *A. muciniphila*-specific primer sequences used were AM1 5'CAG CAC GTG AAG GTG3' and AM2 5'CCT TGC GGT TGG CTT CAG3'. The experimental cycling conditions were: 40 cycles of 50 °C for 2 min, 95 °C for 10 min, 95 °C for 15 s, 60 °C for 1 min.

Because the airway microbiome samples (nasal, oral, and BAL) began with lower biomass starting material, they were subjected to additional quality control using the negative bronchoscope wash to detect possible contamination. The predominant 16S rRNA V4 amplicons of pre-procedure bronchoscope control wash sample (while of low read count) aligned predominantly (84%) to the families Verrucomicrobiaceae and Enterobacteraceae. These families were assumed to be contaminants and were subtracted from the analysis of the airway microbiome samples.

**Animal models**. Both acute and chronic models of airway inflammation were performed. Female and male wild-type and MyD88$^{−/−}$ BALB/c mice aged 6–8 weeks were obtained from Charles River (Sulzfeld, Germany) and housed at

AO Research Institute Davos for the acute ovalbumin (OVA) and acute house dust mite (HDM) models. Mice were housed 4–6 animals per cage in individually ventilated cages in a 12/12 h light/dark cycle with food and water available *ad libitum*. All experimental procedures were carried out in accordance with Swiss law and ethical approval was obtained from Amt fur Lebensmittelsicherheit und Tiergesundheit, Chur. Female wild-type BALB/c mice for the chronic HDM model were bred and housed at GlaxoSmithKline (GSK), Stevenage. Animal studies carried out at GSK were ethically reviewed by the GSK ethics committee and studies were carried out in accordance with Animals (Scientific Procedures) Act 1986 and the GSK Policy on the Care, Welfare and Treatment of Animals. All studies were performed using littermates.

Either ovalbumin (OVA) or house dust mite (HDM) extract were used in acute models of airway inflammation. In the OVA model, mice were sensitized by intraperitoneal (i.p.) injection of 20 µg of OVA grade VI (Sigma-Aldrich, Buchs, Switzerland) emulsified in 500 µg Alum (Pierce, Rockford, IL, USA) in 200 µl sterile 0.9% isotonic Sodium chloride (NaCl) on days 0, 7, and 21, followed by 20 min 1% OVA grade V (Sigma-Aldrich, Buchs, Switzerland) aerosol exposures on days 26, 27, and 28. Negative control animals received NaCl and alum injections and were exposed to the OVA aerosolization. Analysis of mice occurred 24 h after the last aerosol challenge. In the acute HDM model, HDM extract (Greer labs, USA) was administered intra-nasally (i.n.) on day 0 (1 µg), followed by higher dose i.n. administrations on days 7, 8, 9, 10, and 11 (10 µg each day). Negative control animals received saline i.n. on the same days as the positive control animals received HDM extracts i.n. Animals were euthanized on day 12 for analysis. In both acute models, *A. muciniphila* and media control was administered daily (~1 × 10$^8$ cells per dose) by oral gavage, beginning at day −5 until the end of the study. *A. muciniphila* was pre-grown in fresh cultures every day in 10 ml aliquots of anaerobic Mucin v3 media (10% inoculation) at 37 °C. Nitrogen (and boiling) was used to remove the presence of oxygen from Mucin v3 media. Mucin v3 media contains peptone (Fluka), yeast extract (Roth), KH$_2$PO$_4$ (Fluka), NaCl (Fluka), (NH$_4$)SO$_4$ (Acros organics), MgSO$_4$ (Acros organics), CaCl$_2$ (Acros), NaHCO$_3$ (Fluka), D-glucose (Fischer chemical), porcine mucin type II (Sigma), porcine hemin (Acros organics), L-cysteine (Sigma), and water. For the indicated experiments, freshly cultured *A. muciniphila* was heat killed by boiling at 100 °C for 15 min. Cell free supernatants were generated by filtering the culture broth, following *A. muciniphila* 24 h culture at 37 °C, through a 0.22 µm syringe filter (Sarstedt). Two hundred microliter of the filtered supernatants were orally gavaged daily to the mice. Fecal and BAL levels of *A. muciniphila* were determined using quantitative PCR. Identical primers (i.e., AM1 5'CAG CAC GTG AAG GTG3' and AM2 5'CCT TGC GGT TGG CTT CAG3') and cycling conditions were used as described above for *A. muciniphila* quantification in human samples.

BALs were performed with 1 ml of PBS containing 1× protease inhibitor cocktail (Roche, Mannheim, Germany). Red blood cells were removed using red blood cell lysis buffer (Sigma-Aldrich). The total number of leukocytes was counted with a Neubauer counting chamber. For differential cell counts, cytospin preparations were fixed and stained with Diff-Quick (Merz & Dade Ag, Dudingen, Switzerland). Neutrophils, eosinophils, lymphocytes and macrophages were identified using standard morphologic criteria and at least 200 cells were counted per cytospin preparation[54]. Single cell suspensions were obtained from lung tissue using GentleMACS (Miltenyi Biotec). Lung-derived single-cell suspensions were plated at a concentration of 1 × 10$^6$ cells/ml in complete RPMI (Sigma-Aldrich) and were re-stimulated with 50 µg/ml grade VI OVA (Sigma-Aldrich) for 48 h at 37 °C. Culture supernatants were assayed for cytokine levels by Bio-Plex Multiplex Immunoassay System (Bio-Rad). For lung function measurements, mice were intubated under anesthesia and airway resistance was assessed using the forced oscillation technique (FlexiVent system, SCIREQ). Airway resistance was measured in response to increasing concentrations of methacholine from 0, 0.1, 0.3, 1, 3, 10, to 30 mg/ml (Sigma-Aldrich)[55].

For the chronic HDM models, female BALB/c mice were challenged i.n. with either sterile saline or 25 µg of HDM extract for 5 days a week, over 3 weeks (sensitization period, days 1–19). The resulting pulmonary inflammation in the HDM sensitized mice was then allowed to resolve for a period of 2 weeks (days 20 to 33). At the end of the resolution period (day 34), mice were intranasally challenged with either saline or rechallenged with a high dose of HDM (100 µg). HDM rechallenge in this model has been shown to induce granulocytic and lymphocytic infiltration into the airways over a 7 day period (days 34 to 40). Mice were orally gavaged with either mucin v3 media control or *A. muciniphila*, once a day, starting on day 20 (post cessation of HDM sensitization). Oral dosing continued throughout the resolution period and post-HDM rechallenge until day 39. Groups of mice were sacrificed at pre-determined time-points during the resolution phase of the experiment (day 28) and post HDM rechallenge at 4 h (day 34), 24 h (day 35), and 7 (day 40).

**Flow cytometry**. Cells present in the broncho-alveolar lavage fluid (BALF) and lung tissue were quantified using multi-color flow cytometry. Analysis of myeloid and lymphoid subsets was performed on a FACSCantoII machine (Becton Dickinson) running FACSDiva software version 8. Data analysis was performed using FlowJo version 7.6.5 (FlowJo LLC, US). Red blood cells of single cell suspensions containing 1 × 10$^6$ cells were lysed using a 1/10 dilution in sterile distilled water of IO Test 3 solution (Beckman Coulter, UK) for 5 min at room temperature. The

lysing solution was then washed twice by centrifugation at 370×$g$ for 5 min at 4 °C. Supernatants were discarded and cell pellets were resuspended in 20 µl Fc⁻Block$^{TM}$ to block non-specific Fcγ binding sites for 10 min at 4 °C. Without washing off Fc⁻Block$^{TM}$, cells were stained with 50 µl surface marker antibody cocktails for 10 min at 4 °C in the dark after which cells were washed twice to remove any unbound antibodies. Cell pellets were then resuspended in 200 µl of FACS buffer (PBS containing 1% Fetal Calf Serum). For intracellular cytokine staining, freshly isolated cells were stimulated with PMA/Ionomycin (50 ng/ml and 500 ng/ml, respectively) for 4 h at 37 °C and 5% $CO_2$ in the presence of Brefeldin (eBioscience, Vienna, Austria). Cells were permeabilized with reagents from eBioscience (Vienna, Austria). Data was acquired on a calibrated FACSCantoII flow cytometer (Becton Dickinson, UK) using fluorochrome compensated protocols. Gating of intact and single cells (forward scatter Height versus Area) was based on forward and side scatter characteristics. Leukocyte populations were identified as CD45$^+$ in the single cell gate. Cell subsets were identified as lung resident eosinophils (CD11c⁻CD11b$^+$MCHII$^{lo}$CD24$^+$Siglec-F$^{int}$), inflammatory eosinophils (CD11c⁻CD11b$^+$MHCII$^{lo}$CD24$^+$Siglec-F$^{hi}$), monocytes (CD11c⁻CD11b$^+$CD24$^{lo}$MHCII$^{lo}$CD64$^{lo}$Ly6-C$^+$), neutrophils (CD11c⁻CD11b$^+$MHCII$^{lo}$CD24$^+$Ly6-G$^+$), CD8 + T cells (TCRβ$^+$CD8$^+$), B cells (CD45$^+$CD19$^+$), NK cells (CD45$^+$CD49b$^+$), T cell subsets (TCRβ$^+$CD4$^+$CD44$^+$CD62L⁻CD183$^{+/−}$ CD196$^{+/−}$, IL-4$^{+/−}$, IFN-γ$^{+/−}$), or Treg cells (CD3$^+$, CD4$^+$, CD25$^+$, Foxp3$^+$, IL10$^+$). Flow antibodies to CD45 (30-F11, 1/200 dilution), CD183 (CXCR3–173, 1/200 dilution), CD196 (140706, 1/200 dilution), TCRβ (H57–597, 1/200 dilution), CD44 (IM7, 1/200 dilution) and CD62L (MEL-14, 1/200 dilution) were obtained from Becton Dickinson Biosciences, UK; Biolegend, UK CD11b (M1/70, 1/200 dilution), CD64 (X54–5/7.1, 1/200 dilution), MHCII (M5/114.15–2, 1/200 dilution), CD11c (N418, 1/400 dilution), Ly6-G (1A8, 1/400 dilution), Ly6-C (HK1.4, 1/400 dilution), CD8 (53–6.7, 1/200 dilution), CD49b (DX5, 1/200 dilution), CD3 (145–2C11, 1/100 dilution) and CD4 (RM4–5, 1/200 dilution) were obtained from Biolegend, UK; eBioscience, Austria CD24 (M1/69, 1/200 dilution), CD25 (PC61.5, 1/200 dilution), Foxp3 (FJK-16s, 1/50 dilution), IL-10 (JES5–16E3, 1/200 dilution), IL-4 (11B11, 1/100 dilution), and IFN-γ (XMG1.2, 1/200 dilution) were obtained from eBioscience, Austria; Siglec⁻F (ES22–10DB, 1/50 dilution) and CD19 (6D5, 1/30 dilution) were obtained from Miltenyi Biotech, UK.

**Statistical analysis**. 16S rRNA gene sequencing data analyses were performed using QIIME or the R language and environment (version 3.3.2). Samples were rarefied to 24,725 reads, which corresponded to the minimum number of aligned reads to a sample passing quality standards. The Shannon diversity, inverse Simpson, or Chao1 indices and weighted and unweighted UniFrac distances were computed at this rarefaction level. Differential abundance for comparisons of taxa between groups was performed with a one-way ANOVA after correcting for known covariates, age and gender and a 10% FDR correction for comparisons of taxa, which were present in at least 10% of all samples for that body site.

Normalization and differential expression analysis of the transcriptomics data was performed with DESeq2 (version 1.12.4)[56] and custom R scripts (version 3.3.2). Differentially expressed transcripts were identified using a one-way ANOVA, correcting for age, gender and collection site, followed by Tukey's multiple comparison test to compare each group to the non-obese non-asthmatic group. Significance thresholds were set at a $p$-value equal to or less than 0.01 and at least a 1.2-fold change difference. Pathway analysis was performed using MetaCore software (Thomson Reuters, Rochester, NY).

Unless otherwise indicated, data are presented as box-and-whisker plots with the median value and 10–90 percentile illustrated. For analysis of more than two groups, statistical significance was determined using one-way ANOVA and Tukey's multiple comparison test.

## Data availability

Microarray data are deposited in National Center for Biotechnology Information (NCBI) Gene Expression Omnibus (GEO) and are accessible through GEO Series accession number GSE110551. Study gene sequence data and microbiota sequence data are deposited in the NCBI Sequence Read Archive under accession number PRJNA434133. The source data underlying Figs. 1a, 2a, 2b, 4b, 4c, 5b, 5c, 6c–i, 7b, d, e are provided as a Source Data file. All other data, resources and reagents are available from the corresponding author upon reasonable requests.

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

## Acknowledgements

This work has been funded by a research grant from GSK, Swiss National Science Foundation grants (project numbers CRSII3_154488, 310030–144219, 310030–127356, and 310030_144219), Polish National Science Centre grants (project numbers 2011/01/B/NZ6/01872 and 2012/04/M/NZ6/00355), Allergiestiftung Ulrich Müller-Gierok, and Christine Kühne–Center for Allergy Research and Education (CK-CARE).

## Author contributions

D.Mi., K.A.S., K.D.S., M.J., C.A.A., E.M.H. and L.O'M conceived, planned, and oversaw the studies. N.R-P., S.S., M.P., S.U., S.V.H., M.S., A.E., B. P., W.B., M.K-L, P.S-G, R.Fe., R.Fr., N.S., M.K., P.G., K.B.G. and K.L.B. performed laboratory experiments or clinical sampling. C.A., D.Ma., J.R.B., and L.O'M. performed data analysis. D.Mi., D. Ma, K.D.S., J.R.B, C.A.A., E.M.H. and L.O'M. wrote the paper. All authors contributed to reviewing the paper and all authors agreed the final version for submission.

## Competing interests

D.M., D.M., S.U., S.V.H., K.A.S., K.D.S., J.R.B., and E.M.H. are full-time employees of GSK and hold company stock. L.O'M. has consulted for Alimentary Health Ltd. and has received research funding from GSK. C.A.A. has received research support from Novartis and Stallergenes and consulted for Actellion, Aventis, and Allergopharma. M.J. is a consultant to Allergopharma, GER, Anergis, CH, Biomay and received lecture fees from GSK, Allergopharma, Stallergens.

## Additional information

**Peer Review Information** *Nature Communications* thanks Jakob Stokholm and other anonymous reviewer(s) for their contribution to the peer review of this work. Peer reviewer reports are available.

