## [Peer Review File · Nature Communications]

Reviewers' comments:

Reviewer #1 (Remarks to the Author):

In this manuscript Michalovich et al describe a patient cohort containing non-obese asthmatic and obese asthmatic individuals and find a number of differences in inflammatory parameters in the lung as well as differences in their microbiota at different sites of the body. In these results they also find a significant reduction in the proportion of Akkermansia spp in severe compared to mild asthmatic individuals in the intestinal microbiota and thereafter show that oral gavage of Akkermansia muciphila can reduce various features of lung inflammation in different models of experimental asthma in mice a time-dependent manner.

While differences in the microbiota in obesity and asthma have been shown before, this study compares, for the first time, obese asthmatic and lean asthmatic patients side by side in a quite comprehensive manner with regards to their microbiota as well their systemic and lung inflammation. This dataset by itself is a valuable contribution to the field.

However, the study in its current form lacks mechanistic insight and needs considerable improvement. There a number of major points which need to be addressed:

Major points:

1. The authors describe the effect of obesity and asthma to be additive with regards to the inflammatory parameters they measure. However, while these two phenomena strongly potentiate each other with regard to systemic inflammatory parameters, in BAL and lung biopsies the picture becomes much less clear and varies greatly between the different parameters. The authors need to therefore thoroughly discuss this discrepancy between local and systemic inflammatory effects.
2. It is unclear why after focusing the entire study on the relationship between asthma and obesity the authors decide to focus on different severities of asthma towards the end of the manuscript. Furthermore, it is unclear based on the current version of the manuscript what the proportion of obese individuals is in the group suffering of severe asthma. This could potentially completely skew any differential analysis and therefore needs to be accounted for where such comparisons are shown.
3. Exclusion of the genus Verromicrobia from the analysis of the BAL microbiota analysis due the presence in the negative control is peculiar and none convincing. The authors need to improve their low biomass protocol to allow for measuring Verromicrobia without compromising the overall validity of their analysis.

4. It remains unclear in the AM gavage model whether it also leads to Akkermansia colonization of the lung, especially due to the very high bacterial numbers used in this manuscript. The authors need to therefore clarify this point by looking at lung microbiota samples in mice. Furthermore, the authors need to provide evidence that Akkermansia actually colonizes the intestine in their model as the flora appears to be intact upon oral gavage.

5. In addition, it needs to be demonstrated that Akkermansia has a specific effect on lung inflammation by repeating the same set of experiments with at least one but ideally a few other bacteria that did not show any changes in abundance between mild and severe asthma patients. Otherwise the effects observed could be simply due to conserved microbial products and not to an Akkermansia effect.

6. Importantly, more mechanistic evidence is necessary to demonstrate on how Akkermansia affects lung inflammation via oral gavage. Is it a systemic metabolite? The authors need to provide substantially more evidence about the underlying molecular mechanism. While they do show with the Myd88 KO that some TLRs are dispensable for this effect, in these mice some TLR functionality still remains and a more comprehensive model such as Myd88-TRIF double knockout animals would have to be employed to be certain that such factors do not play a role.

7. Furthermore, it is absolutely crucial that these experiments be done in animals which are littermates to avoid microbiota differences causing wrongly interpreted phenotypes.

8. In general, it would be imperative to transfer Akkermansia to germ-free or at least to antibiotic-treated animals and corroborate the reported effects seen in the experimental asthma models.

9. In general, a more comprehensive analysis of the phenotype must be conducted in the experimental asthma mouse models employed to study the effects of Akkermansia. This includes histology of lung structural changes with a quantification thereof to validate the chronic asthma model. Furthermore, analysis in such model often includes a more in depth analysis of the cytokine production of CD4 T cells by flow cytometry.

Minor points

1. The figure legends of most figures do not contain sufficient detail to understand them without the text. They need to include vital information to understand how the experiments were conducted such as the number of samples per group used, the statistical tests employed etc.

2. In Figure 2 it needs to be described how the subset of patients was selected for further analysis and how the inflammatory cells were identified in humans in this case. Furthermore, in Fig2c a quantification of histological differences needs to be provided to validate the claims in a statistical manner
3. In Fig3A it is unclear whether the samples were pooled from all conditions. The authors need to add this information to the figure legend
4. Figure 6e: The legend within the figure is missing
5. For all experimental asthma models schemes of the administration of allergens and bacteria should be included in the main figures to facilitate understanding of the experiments
6. Figure 7: How cells were identified using flow cytometry should be added to the figure legend

Reviewer #2 (Remarks to the Author):

Review: Obesity and Asthma Magnify Disturbed Microbiome-Immune Interactions

Michalovich and coauthors describe immunological and microbiome differences between adults with and without asthma and compare according to obesity or not. They find elevated proinflammatory signatures in peripheral blood and lung in obese participants further increased in asthmatics. Also the microbiota composition associated with both lung and systemic inflammatory profiles. Akkermansia supplementation in murine models seem to alleviate risk.

I would like to congratulate the authors on a massive piece of work and a well-designed study. It is overall an extremely ambitious manuscript, comparing both asthmatics with non-asthmatics and furthermore based on their obesity state, for multiple samples both inflammatory profiles in peripheral blood and lung and also the microbiome in lung, nose, mouth and gut – and in the end also different murine models. Even though the authors do their best to guide the reader through this load of data, the amount of information presented and tests being performed leaves the reader with a feeling of, what was actually take-home messages and what was important. This really hampers the manuscript's readability and might to some degree make it miss the aim.

The 4 clinical phenotypes (obese/non-obese +/- asthma) are included in all analyses, which makes the whole report a bit hard to read, as you need to be very aware of what is purely found in asthmatics and what is obesity and what is associated with both. Also, the asthmatics are in some later analyses further sub-stratified according to severity. I would suggest to do the manuscript

according to asthma-status and then in sub-analyses stratify or adjust for obesity (or BMI – as this is a continuum and inflammation levels would probably be higher with higher BMI).

A very important point is that the authors associate obesity and asthmatic status with elevated proinflammatory signatures in peripheral blood and lung, they find associations between obesity and asthmatic status and microbiota, and they find associations between microbiota and inflammation, but they do not show the combined group of taxa associated with asthma/obesity (or maybe a combined abundance score of these) are associated with inflammatory profiles.

I would like the N of the groups to be stated through every results section and for all figures.

The abstract lacks a description of the population where the study is carried out, numbers and age would indeed be the bare minimum for the reader to get a feeling of the study setting.

Are inflammation and microbiota differences cause or just consequences? Though the authors provide a demographics table, both obesity and asthma in adults are typically associated with many lifestyle factors not described in table 1. So a main question in this regard is whether lifestyle alone can explain some of the observed differences in both inflammation and microbiota? Information on lifestyle factors such as diet, social status, living conditions etc. would be nice additional covariates, which could be included in the models. Also it was not clear to me, whether all analyses were adjusted for significant table 1 differences and supplementary table comorbidities?

The Venn diagrams for the figures 1+2+3 are not very intuitive to understand. What are the circles actually illustrating? The Figure 1a text says biomarker overlap between disease? This should be illustrated in another way. Also in figure 1b, the levels of the p-values illustrated in bars as if the bar length is the important factor. I would much rather see the levels compared and p-values marked in the plots. The same goes for figure 2d+2e and also some figures in the supplement.

They state that they are examining obesity/asthma interactions, and while they demonstrate group differences, they don't really show any statistically interactions by including both phenotypes in the models as interaction-terms.

For the microbiome analysis in fig3a; a 3D plot presented in 2D is impossible to evaluate differences on. I would rather show the 2 main axes, clearly there are differences between the microbial compartments.

What microbial compartment was the random forest classifier build on? Line 126. The text doesn't state this. Also the distribution of relative abundance in y-axis of the 2 scatterplots (fig3b) are clearly not normally distributed, which makes a linear model (and the illustration) a bit misleading, it seems maybe to be driven by outliers.

Instead of the Venn diagrams in figure 3C, I would rather see the actual proportional abundances of the associated taxa, which would also show whether the significant taxa are common or low abundant. Also, differential abundance analyses methods in the microbiome field has been a topic of high interest of many, and good benchmark studies exists. The ANOVA analyses performed here might not be the best suited for the data, as high sparsity is always a problem in these kind of data, and where so many genera are compared. Many DA methods give false results. Methods such as MetagenomeSeq feature model could perhaps be used instead or other methods from the field.

How was the gut enterotypes defined? I cannot find, whether they were done by PAM clustering or by other techniques?

Making a predictive random forest classifier was good for BMI but quite bad for asthma. Selecting Akkermansia in gut, which were associated with asthma status to go on in the murine experiments seems a bit like a random draw, as the authors show many differences between taxa between asthmatics and non-asthmatics in multiple compartments in the DA tests. Here they use the asthma-severity as endpoint without obesity included, which again could argue for making it a more strict asthma manuscript instead of asthma-obesity.

I really like that the experiments also include a chronic allergic model, as this would most likely resemble the cases the study population includes.

Thank you for letting me review this nice manuscript.

Responses to Reviewers' Comments:

Reviewer #1:

In this manuscript Michalovich et al describe a patient cohort containing non-obese asthmatic and obese asthmatic individuals and find a number of differences in inflammatory parameters in the lung as well as differences in their microbiota at different sites of the body. In these results they also find a significant reduction in the proportion of Akkermansia spp in severe compared to mild asthmatic individuals in the intestinal microbiota and thereafter show that oral gavage of Akkermansia municipihila can reduce various features of lung inflammation in different models of experimental asthma in mice a time-dependent manner.

While differences in the microbiota in obesity and asthma have been shown before, this study compares, for the first time, obese asthmatic and lean asthmatic patients side by side in a quite comprehensive manner with regards to their microbiota as well their systemic and lung inflammation. This dataset by itself is a valuable contribution to the field.

However, the study in its current form lacks mechanistic insight and needs considerable improvement. There a number of major points which need to be addressed:

Major points:

- 1. The authors describe the effect of obesity and asthma to be additive with regards to the inflammatory parameters they measure. However, while these two phenomena strongly potentiate each other with regard to systemic inflammatory parameters, in BAL and lung biopsies the picture becomes much less clear and varies greatly between the different*

parameters. The authors need to therefore thoroughly discuss this discrepancy between local and systemic inflammatory effects.

Response: The reviewer is correct in that we noted quantitative additive changes in peripheral blood but did not observe similar clear additive effects in BALs or biopsies. However, we did observe qualitative additive effects as gene ontologies related to both asthma and obesity were observed for obese asthmatics. We have updated Fig. 2c and Fig 2d to show a more expanded list of the enriched pathways, which better illustrates this mixed response within the BALs and biopsies.

We believe that this is an important point to highlight in the manuscript so we have also added the following paragraph to the discussion section (line 232-245):

“The combination of obesity and asthma had significant effects on host inflammatory and transcriptomic responses. This was clearly observed in the peripheral blood transcriptome, where enrichments in ontologies related to inflammatory and innate immune responses were accentuated in obese asthmatics, suggesting an additive effect between obesity and asthma. Within BALs and lung biopsies, non-obese asthma patients showed enrichments for TH2 and asthma-related ontologies, while obese non-asthma patients showed enrichments for tissue-remodeling-related and inflammation-related ontologies. Obese asthmatics displayed enrichments in tissue remodeling-related, inflammation-related and TH2-related ontologies. However, the quantitative additive effect observed for specific ontology enrichments in peripheral blood was not observed for BALs or lung biopsies, rather the obese asthmatic displayed qualitative additive effects in gene ontologies representing both asthma and obesity signatures within the lung. This difference

between systemic versus pulmonary-specific responses may reflect the contribution of multiple organs (e.g. liver and adipose tissue) to systemic inflammatory gene responses²⁶.”

2. It is unclear why after focusing the entire study on the relationship between asthma and obesity the authors decide to focus on different severities of asthma towards the end of the manuscript. Furthermore, it is unclear based on the current version of the manuscript what the proportion of obese individuals is in the group suffering of severe asthma. This could potentially completely skew any differential analysis and therefore needs to be accounted for where such comparisons are shown.

Response: Thank you for raising this point (as is also raised by Reviewer 2) and we agree that the manuscript has benefited significantly from a substantial rewrite to better introduce the effects of obesity and asthma severity as important factors of asthma heterogeneity that both influence the microbiome. Thus, we have substantially improved the Abstract, Introduction and Discussion sections that hopefully will make our findings better understood and more clear. We have also now changed the title of the manuscript to “Obesity and Disease Severity Magnify Disturbed Microbiome-Immune Interactions in Asthma Patients”. Regarding the query on potential skewing of the data due to more obese patients in the severe group, we show in Fig. 5b that there is a significant reduction in Akkermansia levels in both obese and non-obese patients with severe disease.

3. Exclusion of the genus *Verromicrobia* from the analysis of the BAL microbiota analysis due to the presence in the negative control is peculiar and none convincing. The authors need to improve their low biomass protocol to allow for measuring *Verromicrobia* without compromising the overall validity of their analysis.

Response: This result is not related to a low biomass protocol per se, but rather that sequencing of the bronchoscope pre-wash sample (which will have bacterial DNA) revealed a strong *Verromicrobia* signal. We opted for the conservative approach as described by multiple microbiome best practices (For example: Shankar J. Insights into study design and statistical analyses in translational microbiome studies. *Ann Transl Med.* 2017. Poussin C, et al. Interrogating the microbiome: experimental and computational considerations in support of study reproducibility. *Drug Discov Today.* 2018.) of computationally subtracting taxa found on the pre-wash samples collected from the bronchoscope - *“After sequencing, comparisons of reads from the negative controls with the reads from the study samples provide information on whether reagent and/or processing-related contamination exists in the samples. Even if reads from the negative controls are negligible, the researcher may opt for a conservative route and remove any background signals by subtracting the negative control reads from the reads associated with study samples prior to downstream analysis.”*

We agree that this approach may not be ideal, but we would note that these samples came from a bronchoscope wash rather than a true sterile negative control (which would have pointed to a larger reagent contamination problem). Our reasoning for subtracting the Verromicrobia taxa was that this was the predominant taxa identified in the bronchoscope pre-wash controls and that this taxa (to our knowledge of the literature) was not previously reported to be found in the airway microbiome. To convince ourselves that these bacteria were not present in the airway samples, we performed individual qPCRs (with gBlock-synthesized standards) using *Akkermansia muciniphila* specific primers (the same as that used for Akkermansia-specific quantification in fecal samples). The Akkermansia signal was below the limit of detection in the BALs that we tested. Given these results, we believe that the approach of computationally subtracting the Verromicrobia signal represents the most accurate, albeit conservative, representation of these results.

4. It remains unclear in the AM gavage model whether it also leads to Akkermansia colonization of the lung, especially due to the very high bacterial numbers used in this manuscript. The authors need to therefore clarify this point by looking at lung microbiota samples in mice. Furthermore, the authors need to provide evidence that Akkermansia actually colonizes the intestine in their model as the flora appears to be intact upon oral gavage.

Response: We have inserted the fecal and BAL Akkermansia transit data in a new Fig. 6i. Akkermansia levels significantly increase in fecal samples from animals exposed to the bacterium. However, we could not detect any Akkermansia in the BAL samples from the same animals. This data illustrates that Akkermansia is present at high numbers in the gut of

exposed mice, but not within the lung. We did not look for long term colonisation as this is outside the scope of this study.

5. In addition, it needs to be demonstrated that *Akkermansia* has a specific effect on lung inflammation by repeating the same set of experiments with at least one but ideally a few other bacteria that did not show any changes in abundance between mild and severe asthma patients. Otherwise the effects observed could be simply due to conserved microbial products and not to an *Akkermansia* effect.

Response: The reviewer is correct and we have indeed examined a large number of bacterial strains in these respiratory murine models. The vast majority of bacterial strains that we have examined to date do not have activity in these models. We first published negative results in 2010 (Lyons et al. Bacterial strain-specific induction of Foxp3+ T regulatory cells is protective in murine allergy models. Clin Exp Allergy. 2010 May;40(5):811-9), whereby one Bifidobacterial strain (AH1206) protected against allergic airway inflammation, while another Bifidobacterial strain (AH1205) and a Lactobacillus strain (AH102) did not show protective effects. The relevant figure published in that paper is shown here.

Another more recent example from our lab shows that a gram negative microbe, *E. coli* (Akkermansia is also a gram negative microbe), does not protect in the allergic inflammation model (Barcik et al., Allergy. 2019 May;74(5):899-909.).

Thus, we believe that the protective effect observed in our models is a strain-specific effect and is not due to conserved microbial products. In addition, as suggested by the Reviewer, to rule out that the observed effects could be simply due to conserved microbial products, we performed additional experiments with heat killed Akkermansia or filtered culture supernatants from Akkermansia. The new data is included in Fig.6 and shows that heat killed

Akkermansia or filtered culture supernatants from Akkermansia did not have any activity (described in more detail below).

6. Importantly, more mechanistic evidence is necessary to demonstrate on how Akkermansia affects lung inflammation via oral gavage. Is it a systemic metabolite? The authors need to provide substantially more evidence about the underlying molecular mechanism. While they do show with the Myd88 KO that some TLRs are dispensable for this effect, in these mice some TLR functionality still remains and a more comprehensive model such as Myd88-TRIF double knockout animals would have to be employed to be certain that such factors do not play a role.

Response: We have tested heat killed Akkermansia and cell free supernatants from Akkermansia cultures in the OVA model and these did not show any activity (new Fig. 6c). We have also now included the CD4 T cell phenotypes (as a new Fig. 6d) from these animals, which show reduced levels of IL-4 and IFN-gamma positive cells, associated with increased levels of IL-10⁺Foxp3⁺ double positive lymphocytes, but not in the animals administered the heat killed Akkermansia or its culture supernatant. We have inserted the following sentence in the discussion (lines 285-290) – “The increase in IL-10⁺Foxp3⁺ lymphocytes may also mediate anti-inflammatory effects within the lung and *A. muciniphila* administration has previously been shown to increase Tregs in other murine models³⁷. In addition, heat killed *A. muciniphila* was not effective suggesting that either heat sensitive factors or viable bacteria are required for this protective effect and any metabolites secreted *in vitro* were not sufficient to reduce airway inflammation, at least for the concentrations tested.”

As suggested by the reviewer, we have also restricted our interpretation of the results from the MyD88 KO animals in the discussion section to state (lines 282-285) “The *A. muciniphila* mechanism of action is not MyD88-dependent, but may involve other pattern recognition receptors that are MyD88 independent or previously described mechanisms such as improvement of gut barrier integrity or release of metabolites such as nicotinamide that can have systemic effects^{35,36}.”.

7. Furthermore, it is absolutely crucial that these experiments be done in animals which are littermates to avoid microbiota differences causing wrongly interpreted phenotypes.

Response: We completely agree and we confirm that these experiments were performed using littermates. This has been clarified in the methods section.

8. In general, it would be imperative to transfer Akkermansia to germ-free or at least to antibiotic-treated animals and corroborate the reported effects seen in the experimental asthma models.

Response: Germ free or antibiotic-treated animals are usually only used to transplant an entire microbiome or a consortia of microbes, but are not routinely used to test the influence of individual microbes, especially in models examining host inflammatory responses. For the transplant models, it is desirable to reduce the number of bacteria that inhabit the gut so that some microbes from the newly administered transplant or consortia can occupy the available niche that has been created. However, when testing individual microbes, the use of germ-free or antibiotic-treated animals is usually avoided for a number of reasons. Respiratory allergic inflammation is more severe in germ free animals (e.g. Herbst T et al. Dysregulation of allergic airway inflammation in the absence of microbial colonization. *Am J Respir Crit Care Med.* 2011) and therefore a protective effect due to a newly administered microbe can be exaggerated simply due to activation of the immature host innate and adaptive immune system by the presence of a microbe. In addition, antibiotic-induced disruption or dysbiosis of the pre-existing microbiota will significantly influence host immune responses and any effect seen with a newly administered microbe may be related to the microbial dysbiosis and may not be relevant outside this specific type of dysbiosis. Indeed it has been suggested by others that the germ-free animal is not a suitable model for Akkermansia due to the altered mucins and strongly compromised mucus barrier function prior to microbial colonization that is present within the germ free gut (Derrien M et al. Akkermansia muciniphila and its role in regulating host functions. *Microb Pathog.* 2017). In summary, with respect, we suggest that our results in animals, which are colonised by a conventional microbiota, are more relevant in the context of a complex

human disease and we suggest that our current data and models are more informative than the more artificial germ free or antibiotic treated models where competition for niche and utilisation of substrates is very different.

9. In general, a more comprehensive analysis of the phenotype must be conducted in the experimental asthma mouse models employed to study the effects of Akkermansia. This includes histology of lung structural changes with a quantification thereof to validate the chronic asthma model. Furthermore, analysis in such model often includes a more in depth analysis of the cytokine production of CD4 T cells by flow cytometry.

Response: As described above, we have now included flow cytometry data showing changes in cytokine production by lung associated lymphocytes. In addition, the phenotyping of the CD4+ lymphocyte populations in the chronic HDM model described in Fig. 7 shows that CD183+ T cells (a marker preferentially expressed by Th1 cells), CD196+ T cells (a marker typically expressed by Th17 cells) and CD183-/CD196- T cells (a phenotype characteristic of Th2 cells) are all decreased following Akkermansia exposure, which suggests a general anti-inflammatory effect that is consistent with the findings in our other models. Histology was performed initially to validate that structural changes did occur in our models, but we did not evaluate the effect of Akkermansia using histology. Instead of histology, we examined airway hyper-responsiveness to methacholine, which is one of the most robust measures of airway dysfunction and structural changes and we have convincingly shown that Akkermansia reduces AHR in Fig. 6f.

Minor points

1. The figure legends of most figures do not contain sufficient detail to understand them without the text. They need to include vital information to understand how the experiments were conducted such as the number of samples per group used, the statistical tests employed etc.

Response: Legends have been improved as suggested.

2. In Figure 2 it needs to be described how the subset of patients was selected for further analysis and how the inflammatory cells were identified in humans in this case. Furthermore, in Fig2c a quantification of histological differences needs to be provided to validate the claims in a statistical manner.

Response: We apologise for the confusion due to the wording that we used and have amended the text accordingly (lines 107-109) – “We obtained bronchoalveolar lavage (BAL) fluid and bronchial biopsies from obese asthmatics (n=10), non-obese asthmatics (n=12), obese non-asthmatics (n=11) and their non-obese non-asthmatic healthy counterparts (n=8).”. For the BALs and biopsies, this is the entire group that was recruited and sampled for these body sites (n=41). Regarding the quantification of BAL inflammatory cells, we have included the cytopsin and staining methods now in the Methods section (lines 332-335). Statistical analysis of the BAL cytopsin is presented using the appropriate statistics in the results section (lines 113-118). “The total number of inflammatory cells (including eosinophils, neutrophils and lymphocytes) in BAL cytopsin was significantly different between the groups (p=0.042, ANOVA). However, while elevated eosinophils, neutrophils

and lymphocytes were observed in specific obese non-asthma, non-obese asthma and obese asthma patients, none of the inflammatory cell types alone were statistically significantly different between the groups (Fig. 2b).”

The number of biopsies we obtained from each volunteer was limited and we prioritised the biopsies for RNAseq analysis to ensure that we had sufficient RNA from all volunteers to perform RNAseq. Thus, not enough biopsies were available to perform histology from all volunteers and that is why we did not perform statistical analysis on the histological findings of the biopsies. The biopsy histology pictures are included for illustrative purposes only, primarily to visualise the eosinophils and neutrophils that are present within these selected biopsies. To avoid any misunderstanding, we have updated the text (lines 118-119) – “The presence of eosinophils and neutrophils was confirmed by H&E staining in available biopsies (Supplementary Figure 4).”. Please note that we have also moved the histology pictures to the supplementary material so that we do not distract from the main findings of the manuscript in Fig. 2.

3. In Fig3A it is unclear whether the samples were pooled from all conditions. The authors need to add this information to the figure legend.

Response: Correct, these are the samples from the 4 body sites from the 41 volunteers that all 4 body sites were sampled. We have updated the legend as suggested.

4. Figure 6e: The legend within the figure is missing

Response: We apologise and all legends have been updated.

5. For all experimental asthma models schemes of the administration of allergens and bacteria should be included in the main figures to facilitate understanding of the experiments

Response: We agree that this is a very good idea that will facilitate the understanding of each murine experiment. We have inserted the 4 different schemes that we used into Fig. 6 or Fig. 7 as appropriate.

6. Figure 7: How cells were identified using flow cytometry should be added to the figure legend

Response: The staining panel for each cell type has been added to the figure legend.

Reviewer #2:

Michalovich and coauthors describe immunological and microbiome differences between adults with and without asthma and compares according to obesity or not. They find elevated proinflammatory signatures in peripheral blood and lung in obese participants further increased in asthmatics. Also the microbiota composition associated with both lung and systemic inflammatory profiles. Akkermansia supplementation in murine models seem to alleviate risk.

I would like to congratulate the authors on a massive piece of work and a well-designed study. It is overall an extremely ambitious manuscript, comparing both asthmatics with non-asthmatics and furthermore based on their obesity state, for multiple samples both inflammatory profiles in peripheral blood and lung and also the microbiome in lung, nose, mouth and gut – and in the end also different murine models. Even though the authors do their best to guide the reader through this load of data, the amount of information presented and tests being performed leaves the reader with a feeling of, what was actually take-home messages and what was important. This really hampers the manuscript's readability and might to some degree make it miss the aim.

Comment: *The 4 clinical phenotypes (obese/non-obese +/- asthma) are included in all analyses, which makes the whole report a bit hard to read, as you need to be very aware of what is purely found in asthmatics and what is obesity and what is associated with both. Also, the asthmatics are in some later analyses further sub-stratified according to severity. I would suggest to do the manuscript according to asthma-status and then in sub-analyses stratify or adjust for obesity (or BMI – as this is a continuum and inflammation levels would probably be higher with higher BMI).*

Response: This is an important point and we have looked at presenting this data similarly to what is currently suggested by the reviewer. However we did encounter difficulties presenting the data as asthma versus non-asthma, followed by an obesity stratification as we needed significantly more figures to illustrate that many of the inflammatory changes observed in obese asthmatics were not related to their asthma status, but were actually related to their obesity status. In addition, it was difficult to visualise the additive effect of obesity and asthma on inflammatory responses in obese asthmatics without showing the data separately for non-obese asthmatics and obese non-asthmatics. Furthermore, using BMI itself as a continuous variable in the models was complicated by the fact that we recruited volunteers with a BMI of 20-25 (non-obese) or a BMI greater than 30 (obese) and we excluded anyone with a BMI of 25-30. To use BMI in the way suggested would need volunteers in the BMI 25-30 range to be included, which we did not do as we wished to evaluate truly obese rather than overweight individuals. Within the non-obese group, going from BMI 20 to BMI 25 had no effect on inflammatory parameters. Within the obese groups, there were weak correlations between BMI and inflammatory mediators (we show one example below), as it seems that once someone becomes obese then the substantial changes in inflammatory mediator levels has already happened and a further increase in

BMI does not further increase many of these mediators significantly, at least not in a linear fashion. This suggests that there is a “tipping point” relationship rather than a linear dose-response relationship between BMI and inflammation. Given the improvements that we have made to the manuscript text and figures (in line with your other suggestions), we hope that we have improved the readability and flow of the manuscript. However, we feel that we have to show the data as 4 groups in order to provide the reader with the most transparent representations of the data.

Comment: A very important point is that the authors associate obesity and asthmatic status with elevated proinflammatory signatures in peripheral blood and lung, they find associations between obesity and asthmatic status and microbiota, and they find associations between microbiota and inflammation, but they do not show the combined group of taxa associated with asthma/obesity (or maybe a combined abundance score of these) are associated with inflammatory profiles.

Response: We have presented the correlations between individual genera and individual cytokines in supplementary figures 7 and 8, and the correlations between individual genera and inflammatory cells in supplementary table 3. Combining all the significant taxa differences into one abundance score was not helpful to this analysis as some taxa increase while others decrease in relative abundance. Therefore we separated the decreased taxa into one abundance score. Interestingly, when we combined all the significantly reduced genera for the asthmatic lung into one relative abundance score, we indeed found that there was a significant association with BAL IL-5, IL-15 and IFN-gamma levels, but not with any of the other BAL mediators tested. Thank you for this suggestion and we have included this data as a new Fig. 4c.

The Fig 3 legend has been updated – “c The combined relative abundance for taxa reduced in the asthmatic lung (*Butyricimonas*, *Moraxella*, *Propionibacterium*, *Pasteurellaceae*, *Campylobacter*, *Faecalibacterium*, *Bacteroides*, *Parabacteroides*, *Mitsuokella*, *Megasphaera*, *Ruminococcaceae*, *Paraprevotella*, *Phascolarctobacterium*, *Roseburia*, *Enterococcus*, *Bifidobacterium*, *Aeromonadaceae* and *Coriobacteriaceae*) negatively correlated with BAL IL-

5, IFN- γ and IL-15 levels (n=40, all groups combined) using linear regression analysis. Source data are provided as Source Data file 3.”

Comment: *I would like the N of the groups to be stated through every results section and for all figures.*

Response: We agree and the results and figure legends have been updated.

Comment: *The abstract lacks a description of the population where the study is carried out, numbers and age would indeed be the bare minimum for the reader to get a feeling of the study setting.*

Response: We have significantly rewritten the abstract to include details on the population studied.

Comment: *Are inflammation and microbiota differences cause or just consequences? Though the authors provide a demographics table, both obesity and asthma in adults are typically associated with many lifestyle factors not described in table 1. So a main question in this regard is whether lifestyle alone can explain some of the observed differences in both inflammation and microbiota? Information on lifestyle factors such as diet, social status, living conditions etc. would be nice additional covariates, which could be included in the models. Also it was not clear to me, whether all analyses were adjusted for significant table 1 differences and supplementary table comorbidities?*

Response: We collected food frequency questionnaires from all subjects but we did not identify significant differences between the groups examined based on these FFQs. We have included the meat consumption frequency and vegetable consumption frequency as examples below. Thus, the frequency of food types consumed were relatively similar between the groups. Presumably the quantity of food consumed differed between the obese and non-obese groups but unfortunately we were not able to collect this data in this study. Indeed, there is no validated method currently available to reliably measure food quantity intake outside an in-patient setting. We have not included the graphs below in the supplementary material associated with this manuscript as we are already at the maximum number of allowable supplementary figures and tables.

We did not collect detailed information on social status or living conditions. However, we know that all subjects lived within urban neighbourhoods and none lived on a farm or in a rural setting, thus the groups are well controlled regarding this environmental factor. We included the major covariates (i.e. age and gender) in the statistical analysis and these did not significantly impact the findings presented (e.g. see below for the influence of age and

gender on the gut microbiome). In the results section, we explored two major variables, smoking status and inhaled corticosteroids, for their effect on oral, nasal, and BAL microbiome, and observed no statistically significant differences between these groups in the microbiome composition at each airway site (PERMANOVA $p > 0.05$ for each). We also controlled for the comorbidities listed in supplementary table 1 in our analyses and these also did not significantly influence our findings (e.g. see figure below for the influence of diabetes and cardiovascular disease on the gut microbiome). We also checked the influence of comorbidities by excluding people with each comorbidity from the data set and we didn't see a significant effect in the primary results reported in this manuscript. We have not included the graphs below in the supplementary material associated with this manuscript as we are already at the maximum number of allowable supplementary figures and tables.

Comment: *The Venn diagrams for the figures 1+2+3 are not very intuitive to understand.*

What are the circles actually illustrating? The Figure 1a text says biomarker overlap between disease? This should be illustrated in another way. Also in figure 1b, the levels of the p-values illustrated in bars as if the bar length is the important factor. I would much rather see the levels compared and p-values marked in the plots. The same goes for figure 2d+2e and also some figures in the supplement.

Response: We agree with the reviewer that for representing the pathway enrichment data it would be informative to display not just the p-value for the hypergeometric statistical test enrichment, but also to display the proportion of genes in the pathway list that were found to be differentially expressed in each set. These figures are now updated. In addition, we have replaced the Venn diagrams with heatmaps to better illustrate the differences between the groups.

Comment: *They state that they are examining obesity/asthma interactions, and while they demonstrate group differences, they don't really show any statistically interactions by including both phenotypes in the models as interaction-terms.*

Response: The reviewer is correct in that the framework for the statistical models used by us in this manuscript treat the subjects as four distinct groups and not two variables that have the ability to interact (i.e. obesity and asthma). In particular, many of our comparisons are designed to test significance compared to the healthy control group. Therefore we have carefully re-reviewed the manuscript in order to remove any reference to potential interactions as our analysis does not allow us to identify these.

Comment: *For the microbiome analysis in fig3a; a 3D plot presented in 2D is impossible to evaluate differences on. I would rather show the 2 main axes, clearly there are differences between the microbial compartments.*

Response: We have amended the plot as suggested by the reviewer.

Comment: *What microbial compartment was the random forest classifier build on? Line 126.*

The text doesn't state this. Also the distribution of relative abundance in y-axis of the 2 scatterplots (fig3b) are clearly not normally distributed, which makes a linear model (and the illustration) a bit misleading, it seems maybe to be driven by outliers.

Response: The random forest classifier was based on the gut microbiome data and this is now clarified in the text. The reviewer is correct that the linear model presented in Fig 3b is not appropriate given the presence of outliers. We have re-evaluated the analysis using statistical tests more appropriate for data that is not normally distributed. While the trend remains evident, the association between these microbes and BMI falls just outside statistical significance, so we have removed these plots from the manuscript.

Comment: *Instead of the Venn diagrams in figure 3C, I would rather see the actual proportional abundances of the associated taxa, which would also show whether the significant taxa are common or low abundant. Also, differential abundance analyses methods in the microbiome field has been a topic of high interest of many, and good*

benchmark studies exists. The ANOVA analyses performed here might not be the best suited for the data, as high sparsity is always a problem in these kind of data, and where so many genera are compared. Many DA methods give false results. Methods such as MetagenomeSeq feature model could perhaps be used instead or other methods from the field.

Response: We have removed the Venn diagrams as suggested by the reviewer and replaced them with heatmaps that provide additional information on the level and direction of change for each of the genera for each body site. We have also updated the supplementary table 2 to include the mean relative abundances for each group for each of the significant changes. While we acknowledge that each statistical method has advantages and disadvantages, the major advantage with using our current statistical model is that we can account for covariates, which we cannot do with other types of analyses. We did assess other models, including MetagenomeSeq, but found that they were ill-suited for the comparisons we wished to test and the covariates that we felt were important to include. Thus, we respectfully request that the reviewer allows us to present our findings in this manner for the reasons outlined above.

Comment: *How was the gut enterotypes defined? I cannot find, whether they were done by PAM clustering or by other techniques?*

Response: This was calculated by examining the ratio between Bacteroides:Prevotella. A ratio >2 was defined as E1 (Bacteroides dominated), while a ratio <0.5 was defined as E2 (Prevotella dominated). This information has been included in the revised manuscript. See below for the ratio dot plot. We have not included the graphs below in the supplementary

material associated with this manuscript as we are already at the maximum number of allowable supplementary figures and tables.

This ratio reliably separated the majority of study participants with a clear difference in their Bacteroides and Prevotella relative abundances – see below for the relative abundances for each group. Individuals that did not have a clear separation based on their ratio or based on the relative abundance of each microbe were conservatively coded as Emix.

Comment: *Making a predictive random forest classifier was good for BMI but quite bad for asthma. Selecting Akkermansia in gut, which were associated with asthma status to go on in the murine experiments seems a bit like a random draw, as the authors show many differences between taxa between asthmatics and non-asthmatics in multiple compartments in the DA tests. Here they use the asthma-severity as endpoint without obesity included, which again could argue for making it a more strict asthma manuscript instead of asthma-obesity.*

Response: As mentioned above, we have substantially revised the manuscript to better present our rationale for including severity as an important endpoint for comparisons. We felt that Akkermansia was the most novel finding from a microbiome perspective and therefore, we choose this microbe for further assessments in murine models. We have inserted a sentence to better explain our rationale for choosing Akkermansia in our murine models in the results section (lines 189-191) – “In order to determine if any of the microbial changes described above might play a causal role in influencing respiratory inflammation, we selected *A. muciniphila* as a novel microbe to test further in murine models of allergen-induced respiratory airway disease.”. However the reviewer is correct that many of the microbial differences are interesting and should be examined in future models. We have inserted this sentence in the conclusions to highlight this important point (lines 306-309) “However, many of the other changes in microbiome composition that we identified in this study may also be biologically relevant and will need to be investigated in future studies for their relative contributions to the stratification and selection of patients for specific therapies.”.

Comment: *I really like that the experiments also include a chronic allergic model, as this would most likely resemble the cases the study population includes.*

Thank you for letting me review this nice manuscript.

Response: Thank you for your positive comments and for helping us improve this manuscript.

Reviewers' comments:

Reviewer #1 (Remarks to the Author):

The authors have comprehensively addressed all of my critiques and I am happy to recommend acceptance of this interesting manuscript.

Reviewer #2 (Remarks to the Author):

Thank you for allowing me to review this manuscript once again, I am overall very happy with the revision and extra details provided. The authors have done a good job responding to my comments.

I only have a few minor comments remaining.

I still feel that the Akkermansia part feels detached from the major part of the manuscript, the obesity/asthma part. From the provided main analyses (heatmap and table) it does not seem to differ with either asthma or obesity as compared to healthy. Thereby the natural flow of the manuscript is somewhat interrupted. I would like to see the Akkermansia abundances added to this analysis. Even if not significant it would be nice to perhaps see a higher abundance with asthma status. This would bridge to the severity part better.

The results (figure 5a) that leads to the selection of Akkermansia would be interesting to see on genus level.

The figures gets very crowded after the updates, but I agree that the additional information provided is important. I would suggest to add a title for most of the panels (e.g. the four heatmaps in figure 3 – “lower respiratory tract”, “upper respiratory tract”, etc) to make them more intuitive to read without having to go through figure legend.

I find Figure 2 panels c and d difficult to compare directly between phenotypes. Is it possible to illustrate this in another way, more directly showing which pathways are shared between the phenotypes and which are unique?

I would suggest to log transform the CRP levels in the y-axis in figure 5d – also when testing for differences. And perhaps use a trend test for the association with Akkermansia as the authors claim negative correlations. The ANOVA test used now will treat them as independent groups and not in a ranked order.

Best,

Jakob Stokholm

Responses to Reviewers' Comments:

Reviewer #1:

The authors have comprehensively addressed all of my critiques and I am happy to recommend acceptance of this interesting manuscript.

Response: Thank you for your positive response.

Reviewer #2:

I still feel that the Akkermansia part feels detached from the major part of the manuscript, the obesity/asthma part. From the provided main analyses (heatmap and table) it does not seem to differ with either asthma or obesity as compared to healthy. Thereby the natural flow of the manuscript is somewhat interrupted. I would like to see the Akkermansia abundances added to this analysis. Even if not significant it would be nice to perhaps see a higher abundance with asthma status. This would bridge to the severity part better.

Response: Thank you for your comment and we have now included the Akkermansia relative abundances for each of the groups to the text of the manuscript – lines 183-187 “The reduced level of *A. muciniphila* was specifically associated with severe asthma as the relative abundance of *A. muciniphila* was not significantly different between the asthma or obese groups (non-obese non-asthmatic 0.046+/-0.082, non-obese asthmatics 0.035+/-0.069, obese non-asthmatics 0.026+/-0.028, obese asthmatics 0.046+/-0.076, mean+/- standard deviation, $p>0.05$ ANOVA).”. We would prefer to include this data in the text as both the heatmap and table include only those group differences that are statistically significant.

The results (figure 5a) that leads to the selection of Akkermansia would be interesting to see on genus level.

Response: We have now included a new Supplementary Figure 10, which shows the comparison between severe and non-severe asthmatics at the genus level.

The figures gets very crowded after the updates, but I agree that the additional information provided is important. I would suggest to add a title for most of the panels (e.g. the four heatmaps in figure 3 – “lower respiratory tract”, “ upper respiratory tract”, etc) to make them more intuitive to read without having to go through figure legend.

Response: Thank you for the suggestion and we have added the new panel titles to Fig. 3.

I find Figure 2 panels c and d difficult to compare directly between phenotypes. Is it possible to illustrate this in another way, more directly showing which pathways are shared between the phenotypes and which are unique?

Response: In order to highlight the shared and unique pathways, we have moved the top 10 gene ontology enrichment figures from the previous Figs. 2c and 2d to the new Supplementary Figures 5a and 6a. For the new Figs. 2c and 2d we have selected pathway enrichments that are associated with asthma or those associated with airway remodelling and/or proinflammatory responses. We have also updated the associated results text – lines 120-133 *“Transcriptomic analysis of bronchial biopsies revealed a number of genes and related pathways that were differentially expressed in obese and asthmatic individuals. The top ten most significant gene ontology pathway enrichments for each group are illustrated in Supplementary Figure 5a, while expanded heatmaps of immunologically relevant DEGs are illustrated in Supplementary Figure 5b. Asthma-related gene ontology pathways were enriched in both non-obese asthmatics (n=12) and obese asthmatics (n=10), but not obese*

non-asthmatics (n=11), compared to non-obese non-asthmatic controls (n=8, Fig. 2c). Both obese groups displayed significant enrichments in pathways relating to airway remodeling and inflammatory responses (Fig. 2c). In BALs, the top ten most significant gene ontology pathway enrichments for each group are illustrated in Supplementary Figure 6a, while expanded heatmaps of immunologically relevant DEGs are illustrated in Supplementary Figure 6b. Asthma-related gene ontology pathway enrichments were evident in BALs from non-obese asthmatics, while enrichments in gamma-secretase proteolytic targets, epithelial-to-mesenchymal transition and WNT signaling were observed in both obese groups (Fig. 2d).”.

I would suggest to log transform the CRP levels in the y-axis in figure 5d – also when testing for differences. And perhaps use a trend test for the association with Akkermansia as the authors claim negative correlations. The ANOVA test used now will treat them as independent groups and not in a ranked order.

Response: We have updated the figure as suggested including the log transformation and trend analysis.